# Rubisco deactivation and chloroplast electron transport rates co-limit photosynthesis above optimal leaf temperature in terrestrial plants

Andrew P. Scafaro [1,2] ✉, Bradley C. Posch [3], John R. Evans [1], Graham D. Farquhar [1] & Owen K. Atkin [1,2]

Net photosynthetic $CO_2$ assimilation rate ($A_n$) decreases at leaf temperatures above a relatively mild optimum ($T_{opt}$) in most higher plants. This decline is often attributed to reduced $CO_2$ conductance, increased $CO_2$ loss from photorespiration and respiration, reduced chloroplast electron transport rate ($J$), or deactivation of Ribulose-1,5-bisphosphate Carboxylase Oxygenase (Rubisco). However, it is unclear which of these factors can best predict species independent declines in $A_n$ at high temperature. We show that independent of species, and on a global scale, the observed decline in $A_n$ with rising temperatures can be effectively accounted for by Rubisco deactivation and declines in $J$. Our finding that $A_n$ declines with Rubisco deactivation and $J$ supports a coordinated down-regulation of Rubisco and chloroplast electron transport rates to heat stress. We provide a model that, in the absence of $CO_2$ supply limitations, can predict the response of photosynthesis to short-term increases in leaf temperature.

The rapid rise in leaf temperature during a heatwave has detrimental impacts on plant performance[1]. Photosynthesis (net $CO_2$ assimilation—$A_n$) is particularly susceptible to heat stress, and the temperatures at which $A_n$ decreases are well below those for comparable leaf metabolic processes like respiration[1–3]. The Farquhar, von Caemmerer, and Berry $C_3$ photosynthesis model (FvCB model)[4] is a powerful tool for predicting the response of $A_n$ to environmental perturbations, and for determining what aspects of biochemistry limit photosynthetic rate and capacity. The FvCB model predicts $A_n$ based on the minimal rate generated from Rubisco carboxylation reactions ($A_c$), Ribulose-1,5-bisphosphate (RuBP) regeneration associated with chloroplast electron transport rate ($A_r$), and triose phosphate utilisation ($A_p$)[4,5]. The model accounts for declines in $A_n$ above the temperature optimum of photosynthesis ($T_{opt}$) based on concomitant declines in chloroplast electron transport rates ($J$)[4], which has been linked to heat damage of thylakoid membranes[6,7]. However, other studies suggest that $A_c$ can be a greater contributor than $A_r$ to the loss of $A_n$ when leaf temperatures exceed $T_{opt}$[8]. Deactivation of Rubisco and its impact on $A_c$ has long been suspected of contributing to $A_n$ inhibition above $T_{opt}$[9]. Indeed, an analysis by Crafts-Brandner and Salvucci[10] noted that declines in $A_n$ with leaf heating occur well before expectations based on Rubisco kinetics, and are instead consistent with the temperature dependence of Rubisco deactivation. Further experiments and modelling have identified $A_c$ as the rate limiting step in some instances, while other studies have implicated $A_r$ due to declines in $J$ with rising temperatures[6,8,11,12]. Whether $A_c$ or $A_r$ determines $A_n$ above the $T_{opt}$ is often attributed to interspecific differences or environmental factors such as nitrogen availability, growth temperature, and ambient $CO_2$ concentration[6,12,13].

[1]Division of Plant Sciences, Research School of Biology, The Australian National University, Canberra, ACT 2601, Australia. [2]Centre for Entrepreneurial Agri-Technology, Gould Building, Australian National University, Canberra 2601, Australia. [3]Department of Research, Collections and Conservation, Desert Botanical Garden, Phoenix, AZ, USA. ✉e-mail: andrew.scafaro@anu.edu.au

An alternative possibility is that $A_c$ and $A_r$ are both regulated to be co-limiting. For example, Sage[14] proposed and observed[15] synchronised $A_c$ and $A_r$ biochemical adjustments within minutes of altering irradiation and $CO_2$ concentrations.

Not only can the capacity of Rubisco to fix $CO_2$ and the light dependent generation of RuBP be impaired by heat, but the availability of $CO_2$ substrate at the site of assimilation can fall and become limiting. Reduced $A_n$ due to falling intercellular and chloroplast $CO_2$ concentrations following heat-associated rises in vapor pressure differences between leaves and air have been observed[16–18]. Additionally, foliar $CO_2$ loss from photorespiration and respiration in the light ($R_L$) may contribute substantially to declining $A_n$ under high temperature[19], as both processes rise sharply with warming[20,21]. From a modelling perspective, this means that for each new temperature, several parameters must be treated independently and updated. Despite, or perhaps because of, the above numerous aspects of photosynthetic metabolism that are impaired by heat, it is difficult to establish a general predictor for the decline in $A_n$ at relatively moderate temperature applicable across many higher plants.

The maximum carboxylation capacity of Rubisco ($V_{cmax}$) is a key parameter in the FvCB model[22]. Gas-exchange estimates of $V_{cmax}$ increase exponentially with temperature before peaking and then declining at higher temperatures; the point of decline is influenced by acclimation to growth temperature[23,24]. This decline in apparent $V_{cmax}$ is not explained by susceptibility of Rubisco to high temperature. Rubisco is a relatively thermally stable protein, and in vitro thermal characterisation of Rubisco, in the absence of phosphorylated compounds, demonstrates that it has an exponentially rising carboxylation rate constant ($k_{cat}$) and remains functional at temperatures far exceeding the in vivo deactivation point[25–27]. Rather, the deactivation of Rubisco is due to the heat sensitivity of Rubisco activase (Rca), the accessory protein of Rubisco that removes tightly bound sugar phosphate inhibitors from the Rubisco active site. Rubisco is prone to decarbamylation, where a $Mg^{2+}$ ion and $CO_2$ molecule is not bound to the active site prior to RuBP substrate binding, leading to deactivation and the need for Rca to remove bound RuBP from the active site[28]. Loss of Rca function leads to a reduction in the proportion of Rubisco catalytic sites that are activated and to concomitant declines in photosynthesis[25,26]. Accurately modelling $A_n$ at temperatures above the $T_{opt}$ therefore requires knowledge about the activation state of Rubisco catalytic sites.

A central assumption of the FvCB model is that all Rubisco catalytic sites in a leaf are functional and invariant[29]. However, this assumption is inconsistent with the observed decline in apparent $V_{cmax}$ values (calculated from gas-exchange data) and the number of functional sites under rising leaf temperature. One work-around is to regard $V_{cmax}$ as a variable in time, like temperature itself. Alternatively, a more satisfying reconciliation of this inconsistency is to calculate $V_{cmax}$ based on the $k_{cat}$ of Rubisco and its deactivation based on biochemical observations[14,30]. We hypothesise that we can accurately predict $A_n$ above the $T_{opt}$ by allowing the number of functional Rubisco catalytic sites to vary with temperature. We explore the extent to which this Rubisco deactivation-based $A_c$ corresponds to a previous model that predicts $J$-dependent $A_r$ declines in $A_n$. We tested the scalability of these models against published temperature response curves and a global composite response curve of $A_n$ measured over a wide range of leaf temperatures, biomes, and plant functional types.

## Results

### Accounting for Rubisco deactivation in $C_3$ photosynthesis models

To capture $V_{cmax}$ when accounting for the biochemically reported deactivation of Rubisco with rising temperature, the Sharpe-Schoolfield equation for enzyme deactivation at high temperatures[31] was used:

$$V_{cmax} = \frac{n \cdot k_{cat}}{1 + e^{\left[\frac{E_d}{R}\left(\frac{1}{T_{0.5}} - \frac{1}{T_K}\right)\right]}} \quad (1)$$

The expected number of Rubisco catalytic sites ($n$), and the $k_{cat}$ of Rubisco at a given temperature were based on reported values from more than 70 higher plant species as described in the methods and presented in Supplementary Table 1 and Supplementary Fig. 1. We iteratively solved the deactivation energy ($E_d$) and the temperature at which enzyme activity was halved ($T_{0.5}$)[31] through a non-linear least-squares regression fit of Eq. 1, with the numerator set to unity, to published biochemical responses of the Rubisco activation state to leaf temperature (Fig. 1a). Heat susceptibility of Rca varies depending on acclimation to growth temperature and the thermal environment to which a plant is adapted[32,33]. Therefore, as with Rubisco kinetic parameters, we separated activation data based on whether a species grew at a day temperature below (cool) or above (warm) 25 °C, respectively (Supplementary Table 2). Our analysis included 17 species, consisting of seven cool and 10 warm grown species. Twelve of these species were herbaceous, while only three were temperate trees and zero were tropical trees, highlighting the current lack of knowledge of Rubisco activation across plant functional types. Non-linear least-squares iteration found a cool growth $E_d$ of 199 kJ mol⁻¹ and $T_{0.5}$ of 39.0 °C, and a warm growth $E_d$ of 212 kJ mol⁻¹ and $T_{0.5}$ of 42.4 °C. The decline in $V_{cmax}$ derived from this mechanistic link to Rca functional control of Rubisco activation closely matched the peaked Arrhenius equation with an acclimation term, a model derived from empirical gas-exchange based apparent $V_{cmax}$ data[23] (Fig. 1b).

Without a Rubisco deactivation term and in the absence of photorespiration (i.e. $O_2$ parameterised to zero), modelled $A_c$ did not reach a $T_{opt}$ below 50 °C for either cool or warm grown plants (Fig. 1c). When accounting for photorespiration (i.e. $O_2$ parameterised to atmospheric concentrations of 21%) but not Rubisco deactivation, $A_c$ reached a $T_{opt}$ at the relatively hot temperatures of 45.1 °C for cool and 46.6 °C for warm grown plants. With both photorespiration and Rubisco deactivation accounted for, the $T_{opt}$ of $A_c$ was 29.4 °C for cool and 32.7 °C for warm grown plants and $A_c$ declined sharply as temperatures exceeded these $T_{opt}$ (Fig. 1c). The use of Rubisco kinetics from a variety of studies led to variation in predicted rates of $A_c$, but the general pattern of decline due to Rubisco deactivation remained prominent (Supplementary Fig. 2). We further compared $A_c$ to $A_r$ predictions based on temperature response of $J$ (Eq. 8 and Supplementary Table 1). $J$ and its temperature response were derived from published observations comprising 26 species, 23 of which were either herb/grasses or temperate trees (Supplementary Table 3). $A_r$ which factored in the temperature response of $J$ declined at relatively mild temperatures, with a $T_{opt}$ of 28.1 °C and 31.9 °C for cool and warm grown plants, less than 1.5 °C difference from the corresponding $T_{opt}$ of $A_c$ which included Rubisco deactivation (Fig. 1c).

### Rubisco deactivation and electron transport can predict individual species and the global pattern of declines in $A_n$

To test the accuracy of the model which included Rubisco deactivation, $A_c$ and $A_r$ predictions were compared to observed $A_n$ from 75 published temperature response curves comprising 49 $C_3$ species (Supplementary Table 4). Observations and $A_c$ modelled curves for species representing six major plant functional types demonstrated that high temperature-induced declines in $A_n$ can be accurately predicted when Rubisco deactivation is accounted for (Fig. 2). However, significant inaccuracy was observed when modelling extremophiles. *Deschampsia antarctica*, an Antarctic grass that was grown at 12 °C, had a mean root mean squared error (RMSE) of 6.6 μmol $CO_2$ m⁻² s⁻¹

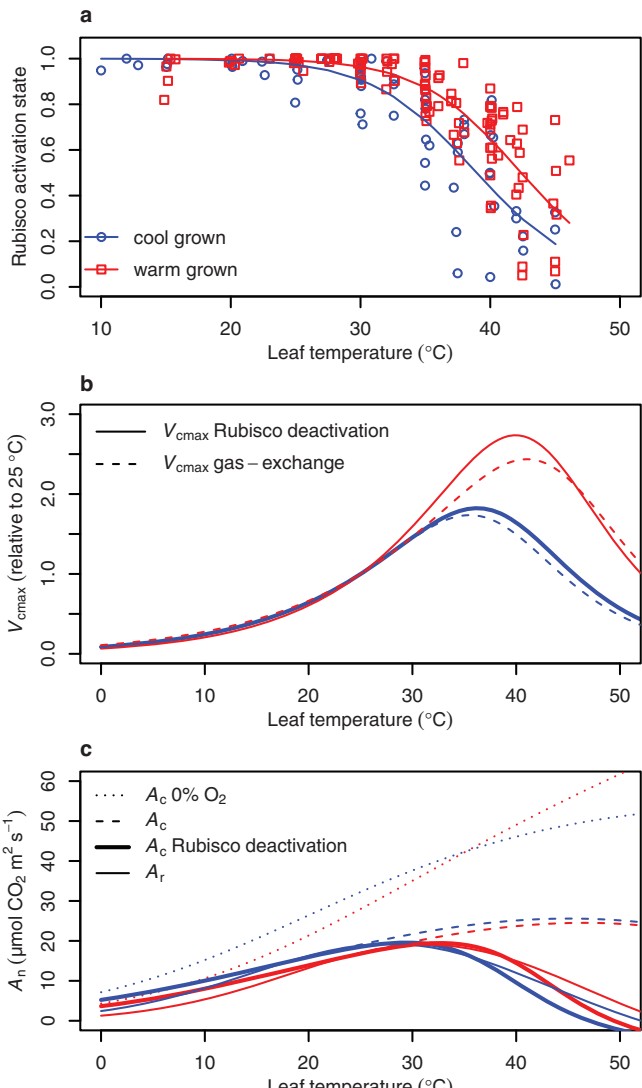

**Fig. 1 | Deactivation of Rubisco and its implications for $V_{cmax}$ and $A_n$. a** A collection of published fractions of total functional Rubisco sites in a leaf (Rubisco activation state−points) at a given temperature (refer to Supplementary Table 2 for metadata). A Sharpe-Schoolfield equation (solid lines) accounting for enzyme deactivation at high temperature (Eq. 1) with the numerator set to unity was fit through non-linear least squares regression for both the cool (grown at <25 °C; blue circles) and warm (grown at >25 °C; red squares) growth datasets. **b** The temperature responses of $V_{cmax}$ that we derived from Rubisco deactivation (solid lines) and the apparent $V_{cmax}$ derived from gas-exchange estimates and an Arrhenius peaked model (dashed lines) with an acclimation parameter set at 24 and 36 °C based on Kattge and Knorr[23]. Cool (blue) or warm (red) grown species dependent on their day growth temperature being below or above 25 °C, respectively. **c** The net photosynthesis $CO_2$ assimilation rate ($A_n$) predicted from carboxylation limited photosynthesis ($A_c$) modelled with no $O_2$ (i.e. no photorespiration) and assuming Rubisco is totally active (dotted lines); or at 21% $O_2$ but assuming Rubisco is totally active (dashed lines), accounting for Rubisco deactivation (bold solid lines), or assuming RuBP regeneration limited photosynthesis ($A_r$) based on chloroplast electron transport ($J$) and its response to temperature (solid lines).

limited difference in predictive power between cool and warm grown species or among plant functional types (Fig. 2c, d). The RMSE between predicted $A_c$ and observed $A_n$ was 2.1 μmol $CO_2$ m$^{-2}$ s$^{-1}$ in cool grown species and 2.7 μmol $CO_2$ m$^{-2}$ s$^{-1}$ in warm grown species. There was no significant difference (Welch Two Sample $t$ test; df = 1124, $t = -0.092$, $p = 0.93$) in model predictions based on $A_c$ that included Rubisco deactivation or based on $A_r$ that included temperature dependence of $J$ (Fig. 3 and Supplementary Fig. 3).

To further test the predictability of the model on a global scale that was independent of species, we compared model predictions with a composite curve comprised of the previously reported $A_n$ temperature response observations relativised to their $T_{opt}$ (Fig. 4a). The predicted fall in $A_c$ with rising temperature and the differences between cool and warm grown plants both closely aligned with observations. A similar prediction resulted based on $A_r$. We further compared the model incorporating Rubisco deactivation with a global composite temperature response curve generated from the mean $A_n$ rates of a dataset totalling 13,876 individual gas-exchange observations from 311 species, representing a wide range of plant functional types[34]. The curve was developed by binning and averaging the observations for each degree of measured leaf temperature. The number of observations per degree was normally distributed, with a peak at 30 °C (Supplementary Fig. 4). Again, we found a close relationship between the observed temperature response of $A_n$ and the model predictions, with a RMSE of 1.7 and 2.0 μmol $CO_2$ m$^{-2}$ s$^{-1}$ for $A_c$ and $A_r$ warm grown predictions applied, respectively (Fig. 4b). Rubisco deactivation and the temperature response of $J$ thus effectively predicted the peak and decline in $A_n$ that occurs with rising leaf temperature on an interspecific level.

## Discussion

Improved understanding of why photosynthesis is impaired by even moderate heat stress is needed if we are to accurately account for the influence of rising atmospheric temperatures on global vegetation. By accounting for the temperature-dependent change in the activation state of Rubisco, we were able to accurately predict warming-induced declines in $A_n$ on an individual species, biome, and global interspecific level. The declines in $A_n$ that were predicted based on $A_c$ and by accounting for Rubisco deactivation were not significantly different from declines predicted by $A_r$ when accounting for the interspecific temperature dependence of $J$. It has recently become apparent that $A_c$ and $A_r$ are optimised to co-limit $A_n$ at a given growth temperature across a wide-range of species[35, 36]. Our results support a continuation of $A_c$ and $A_r$ co-limitation on shorter timeframes of one hour or less when leaf temperatures rise above the $T_{opt}$.

Recent modelling on a broad range of higher plants has pointed towards $T_{opt}$ adapting and acclimating to growth temperature due to photosynthetic biochemistry rather than $CO_2$ conductance limitations[37]. Similarly, by assuming a constant $CO_2$ conductance in our model and thus removing the effect of water stress, we demonstrated that heat stress per se can explain declines in $A_n$ beyond $T_{opt}$. The study by Lin et al.[34], from which we obtained the global composite $A_n$ temperature response curve (Fig. 4b), demonstrated that stomatal conductance is regulated to maximise woody tissue development, including the allowance of greater water loss at warmer growth temperatures in wet environments. There is growing evidence that many plants with access to water keep their stomata open despite high air temperature and vapour pressure deficit as a means of transpirational cooling[38–40]. Therefore, there appear to be instances when plants prioritise thermoregulation over managing drought risks, although whether transpiration can reduce leaf temperatures below that of the surrounding air may be limited in natural sunlit canopies due to biophysical factors such as radiative heating[41]. Although we removed the confounding influence of water stress in this study, water availability remains influential in reducing $A_n$ in species that close their stomata to

due to it maintaining stable $A_n$ to temperatures below 10 °C (Fig. 2a). A similar lack of predictability occurred for *Larrea divaricata*, an arid shrub that was grown at 44 °C, which had a mean RMSE of 8.9 μmol $CO_2$ m$^{-2}$ s$^{-1}$ due to the modelled decline in $A_n$ occurring ~8 °C before the observed decline (Fig. 2b). The lack of predictive power for these extremophiles occurred irrespective of whether predictions were derived from $A_c$ with Rubisco deactivation or $A_r$ models. There was

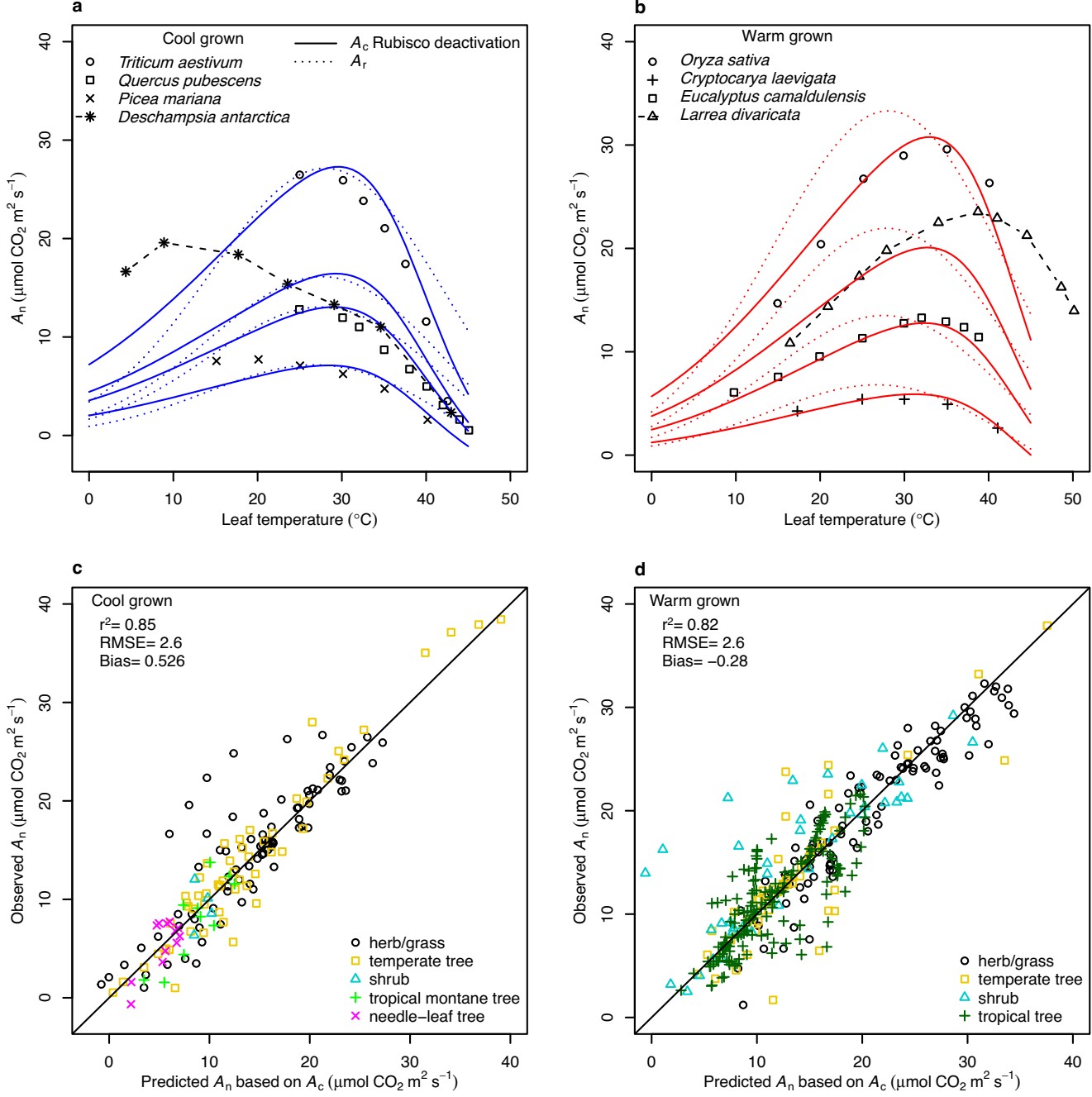

**Fig. 2 | Observations, modelled simulations, and the predictability of the leaf temperature response of net CO₂ assimilation ($A_n$) for individual species and plant functional types. a** The $A_n$ temperature response of four representative cool grown species: *Triticum aestivum* (herb/grass), *Quercus pubescens* (temperate tree), *Picea mariana* (needle-leaf tree), and *Deschampsia antarctica* (herb/grass). For *D. antarctica*, an extremophile that was grown at 12 °C, the broken line between observations highlights the deviation from the model prediction below 30 °C for this species. Points are observations and the curves are the Rubisco carboxylation limited assimilation rates ($A_c$) with Rubisco deactivation included parameterised to cool grown plants (solid blue lines), or RuBP regeneration limited CO₂ assimilation rates ($A_r$) parameterised to cool grown plants (dotted blue lines). **b** Four representative warm grown species: *Oryza sativa* (grass), *Cryptocarya laevigata* (tropical

tree), *Eucalyptus camaldulensis* (temperate tree), and *Larrea divaricata* (shrub). For *L. divaricata*, an extremophile that was grown at 44 °C, the broken line between observations highlights the deviation from the model prediction above 30 °C for this species. Points are observations and the curves are the $A_c$ with Rubisco deactivation included for warm grown plants (solid red lines), or $A_r$ for warm grown plants (dotted red lines). **c, d** Predictions of $A_c$ which included Rubisco deactivation were plotted against corresponding observations for cool (**c**) and warm (**d**) grown plants. The coefficient of determination ($r^2$), a 1:1 ratio (solid line), the root mean squared error (RMSE) between observed and predicted values (µmol m⁻² s⁻¹), and the bias in observations being greater than predictions (µmol m⁻² s⁻¹) for each growth environment are provided. Plant functional types are indicated by differing symbols and colours.

preserve water, particularly during hot and dry conditions when vapour pressure deficit is high and soil moisture is low[16,39]. Indeed, our amended $A_c$ model overestimated the temperature at which $A_n$ began to decline when compared to tropical tree and lianas species in Panama (Supplementary Fig. 5). This was consistent with the published

declines in stomatal conductance and intercellular CO₂ concentrations in response to leaf heating for these same species[17]. Another aspect of CO₂ conductance that can influence $A_n$ is the rate of CO₂ diffusion from intercellular airspaces to the site of chloroplasts, termed mesophyll conductance. Mesophyll conductance appears to either increase or

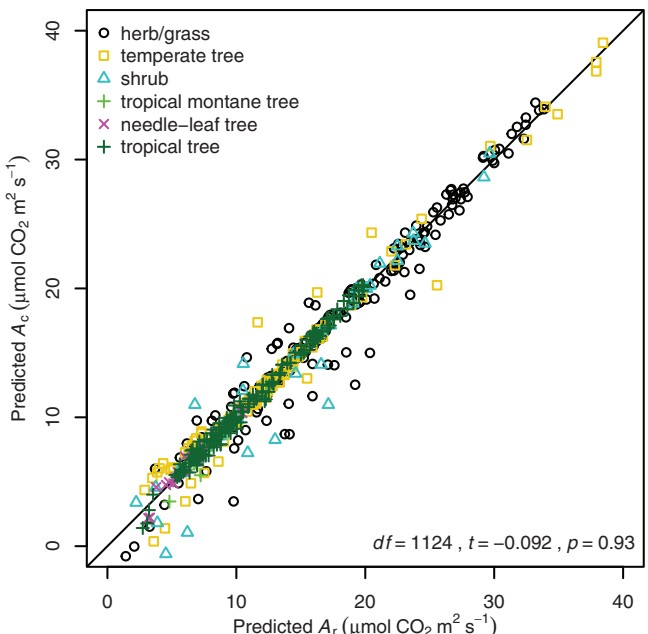

**Fig. 3 | Comparison of net photosynthetic $CO_2$ assimilation predictions based on Rubisco carboxylation that included Rubisco deactivation ($A_c$) and rates of RuBP regeneration ($A_r$).** Values correspond to temperature and species observation predictions for the temperature response curves of 49 species previously published (Supplementary Table 4). Plant functional types are indicated by different symbols and colours. The degrees of freedom (df), t value (t), and p value (p) of a Welch Two Sample t test comparing $A_c$ and $A_r$ are presented in the graph. The solid line is the 1:1 ratio.

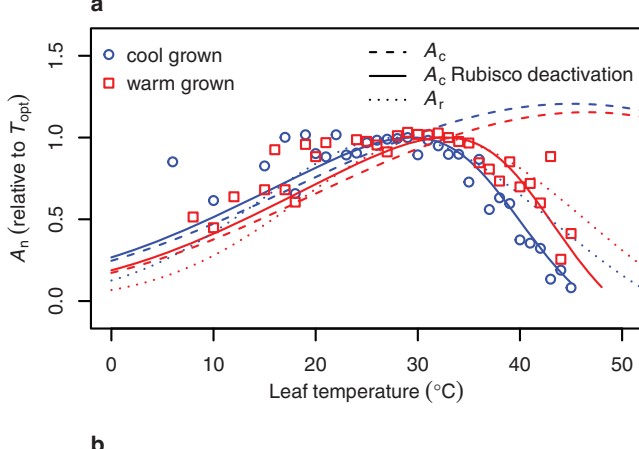

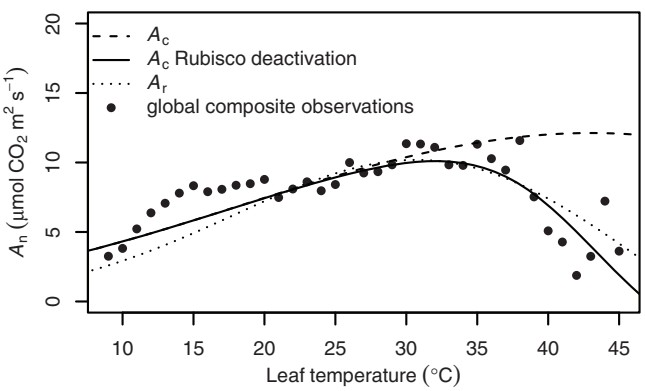

**Fig. 4 | The implications of Rubisco deactivation on net $CO_2$ assimilation predictions. a** Net $CO_2$ assimilation rates ($A_n$) from temperature response curves representing 49 C₃ plant species (Supplementary Table 4). Individual temperature response curves were relativised to the maximum rate achieved at the temperature optimum ($T_{opt}$), and points represent the means of relativised values that were binned by degree Celsius. Observed points were plotted against Rubisco carboxylation limited assimilation ($A_c$) with Rubisco totally active (dashed lines), accounting for Rubisco deactivation (solid lines), or assuming RuBP regeneration limited photosynthesis ($A_r$) based on chloroplast electron transport ($J$) and its response to temperature (dotted lines). Rubisco kinetics were categorised as cool (blue) or warm (red) grown species dependent on their day growth temperature being below or above 25 °C, respectively. **b** A composite temperature response curve (closed circles) based on a global dataset comprising of 311 species and 13,876 observations[34]. Mean $A_n$ values, independent of species and experiment, were binned by degree Celsius. Observed points were plotted against Rubisco carboxylation limited assimilation ($A_c$) with Rubisco totally active (dashed lines), accounting for Rubisco deactivation (solid lines), or assuming RuBP regeneration limited photosynthesis ($A_r$) based on chloroplast electron transport ($J$) and its response to temperature (dotted lines).

remain constant with short-term rises in leaf temperature across many species[42]. This will contribute to intercellular $CO_2$ drawdown and may exacerbate Rubisco $CO_2$ substrate limitations when water is limited and temperatures hot.

Rubisco specificity for $CO_2$ significantly shapes $A_n$ under moderate, sustained warming, and photorespiration becomes a greater contributor to $CO_2$ loss as leaf temperature rises (Fig. 1c). However, photorespiration cannot account for the extent of $A_n$ decline that occurs as leaf temperature exceeds $T_{opt}$ (Fig. 1c). Indeed, model predictions of $A_n$ that account for photorespiration but not for Rubisco deactivation or $J$ limitations far exceed observed $A_n$, estimating a $T_{opt}$ of 45 °C in cool grown plants (Fig. 1c). It is therefore unlikely that Rubisco deactivation and $J$ are downregulated as a mechanism to limit photorespiratory $CO_2$ loss considering both contribute far more to declines in $A_n$ above $T_{opt}$ than photorespiration does, assuming there is no other cost of greater photorespiratory 2-phosphoglycolate metabolism apart from $CO_2$ release. Rather, unavoidable heat damage to membranes and proteins likely set the thermal limits in Rubisco and $J$ capacity.

There is extensive literature linking Rubisco deactivation to the thermolability of Rca[43]. Rubisco deactivation with rising temperature is attributed to loss of Rca, due to the role of Rca in maintaining functional Rubisco catalytic sites and it being characteristically susceptible to degradation under relatively mild heat[44]. Our findings are consistent with the link between Rubisco deactivation and Rca, as our cool grown $T_{0.5}$ of 39.0 °C is close to the 35-38 °C range within which isolated Rca from temperate wheat (*Triticum aestivum*, L.) loses 50% of its functionality[33,45]. Similarly, the warm grown $T_{0.5}$ of 42.4 °C is within the 40 to 43 °C range in which isolated Rca from warm grown rice (*Oryza sativa*, L.) loses 50% of its ability to activate Rubisco[33,46]. The difference in $T_{0.5}$ between cool and warm grown observations in the global activation state data (Fig. 1a) reflects the previously documented pattern of Rca acclimating and adapting to the prevailing growth temperature, including becoming more thermally stable in hotter environments[32,33].

The dynamic and reversible decline in $J$ at high temperatures[47] has been linked to heat susceptibility of thylakoid membranes and their constituents[7,11,48]. The oxygen evolving complex of photosystem II (PSII) and the cytochrome $b_6/f$ complex (Cyt $b_6/f$) seem particularly important in setting dynamic temperature-effected rates of $J$[49–51]. There are four Mn atoms per PSII reaction centre responsible for oxidation of $H_2O$. Mn is held by 33 kDa D1 proteins. Prolonged heat stress can dislodge D1 proteins and subsequently $Mn^{2+}$ ions from the oxygen evolving complex of higher plants, resulting in a decline in $J$[52–54]. The disruption in electron accepting ability of PSII leads to the reaction centre being overly oxidised ($P_{680}^+$) and conducive to ROS formation, which can impair D1 protein synthesis and further diminishes PSII functionality[55]. The heat sensitivity of the PSII oxygen evolving complex makes it a key reason for why $J$ declines under high leaf temperatures. Heat damage to thylakoid membranes is not confined to

PSII. Moderate temperature of 40 °C is shown to disrupt the balance of electron flow between PSII and PSI which is controlled by Cyt $b_6/f$, and through damage or regulation, the flow of electrons through Cyt $b_6/f$ leads to an overreduction of PSI upon heat exposure[56]. With an overly reduced PSI, cyclic electron flow is upregulated as a means of dissipating electrons and preventing irreversibly damage to the stroma[7,56,57]. However, cyclic electron flow is insufficient to maintain $A_n$ in the absence of linear electron flow since it does not produce the necessary NADPH to run the Calvin-Benson cycle[57]. This is a strong indicator that the imbalance in electron flow as temperatures exceed the $T_{opt}$ contributes to declining $J$ and subsequently $A_r$.

The alignment of Rubisco deactivation and declines in $J$ suggest a closely aligned temperature limitation on the functionality of both, a tight temperature dependent regulation of one to a limitation in the other, or a combination of the two. Rca activity is modulated by ATP and inhibited by competitive binding of ADP[58–60]. Declines in $J$ with rising temperature due to electron transport imbalance, leakiness of thylakoid membranes, or other damage likely reduce stromal ATP concentrations. Lower stromal ATP concentrations reduce the active state of Rubisco[61], presumably through reduced Rca activity. Conversely, a lack of $CO_2$ fixation by Rubisco due to heat instability of Rca may lead to an accumulation of RuBP, reductant, and ATP. Recent analysis suggests that Cyt $b_6/f$ tightly controls the dynamic flow of electrons between PSII and PSI, thus an accumulation of reductant and ATP would quickly downregulate Cyt $b_6/f$ electron transfer[51]. Interestingly, Rca has previously been found to associate with thylakoid membranes under heat stress in spinach (*Spinacia oleracea*, *L.*)[62], and a recent report in rice noted a reduction in the quantum yield of photosystem I with overexpression of Rca[63]. Whether Rca and components of the electron transport chain interact directly during heat perturbation to coordinate downregulation of photosynthesis with rising temperature requires further exploration.

Model predictions based on $A_c$ that included Rubisco deactivation and $A_r$ were accurate across a wide range of cool and warm grown higher plant species from a range of plant functional biomes (Figs. 2 and 4). However, the model fits were poor in relation to predicting $A_n$ of plants adapted to extreme cold and heat. The $A_n$ temperature responses of *Deschampsia antarctica*, a small grass native to antarctica, and *Larrea divaricata*, a desert shrub, diverged from model predictions (Fig. 2). This suggests that extremophiles like *D. antarctica* and *L. divaricata* may have unusually cold or heat stable photosynthetic constituents. For example, *L. divaricata* may have a variant of Rca similar to that of the CAM plant *Agave tequilana* which has an Rca isoform that remains active up to 50 °C[32]. Thus, while our model may not be applicable to extremophiles, it may provide a novel means of identifying species with superior thermal stability of photosynthetic components, as indicated by observed $A_n$ of a species far exceeding the model predictions.

In conclusion, we have demonstrated the importance of accounting for Rubisco deactivation when modelling photosynthesis above the $T_{opt}$. By doing so, the model we presented more accurately predicted previously observed declines in $A_n$ with rising leaf temperature across a broad range of higher plant species and functional types. Our predictions of $A_n$ based on Rubisco deactivation are in close agreement with $A_n$ predicted from the temperature dependence of $J$, suggesting both are likely highly coordinated and co-limit photosynthesis as temperatures rise. Attempts to engineer improvement in photosynthesis at high temperature should thus focus on both Rubisco and electron transport characteristics, as a benefit to one without a benefit to the other is likely to result in only incremental improvements in heat tolerance. Although $T_{opt}$ is known to shift with growth temperature at a finer scale than simply below or above 25 °C, the limited number of published Rubisco activation state and $J$ temperature curves prohibits model parameterisation that would allow predictions of finer scale adjustments in $A_n$ to changing growth temperature. Further

studies that characterise the temperature dependence of Rubisco deactivation and temperature dependence of $J$ – ideally from a wide spread of plant functional types – will improve the accuracy of the models we present. Finally, we demonstrated that neither $CO_2$ substrate supply limitation nor photorespiratory $CO_2$ loss was needed to explain high temperature-induced decreases in $A_n$. However, many future heatwaves are likely to coincide with drought, and drought will reduce $CO_2$ conductance and increase photorespiratory $CO_2$ loss, exacerbating the stress caused by Rubisco deactivation and declines in $J$.

## Methods

### Acquisition of Rubisco activation state, electron transport, and $CO_2$ assimilation rate data

Published temperature response curves of Rubisco activation state, chloroplast electron transport rate ($J$), and net $CO_2$ assimilation rate per unit leaf area ($A_n$) for individual species were collated through searching the published scientific literature. Metadata relating to the publication and conditions of measurement are provided in Supplementary Tables 2, 3 and 4. Data were either extracted directly from tables or by recording values from graphs using the data extraction package "digitize-package" in R statistical software[64] following package instructions. For Rubisco activation data, papers that biochemically characterised the Rubisco activation state to increasing leaf measuring temperature were collated. Values were normalised to the maximum recorded value in each temperature curve. Data were separately analysed based on the mean day growth temperature being below (cool) or above (warm) 25 °C. Where plants were grown at 25 °C, cool or warm grown distinctions were made based on whether the peak in photosynthesis was below or above 30 °C, respectively. For $A_n$ observational data, studies that measured $A_n$ on a leaf area basis equal or less than one hour after increasing leaf measuring temperature were included. Where light, $CO_2$ concentrations, or nitrogen were varied, we only used curves with the greatest light or nitrogen application and $CO_2$ concentrations corresponding to ambient concentrations of $400 \pm 50$ μmol mol$^{-1}$. Only curves with more than four temperature data points were included.

The interspecific temperature response curve comprising observations obtained from 310 species (Fig. 4b) was based on a global set of gas-exchange data as presented in Lin et al. (2015)[34]. $A_n$ values and the leaf temperature at which they were recorded were binned per °C and the mean used to generate a global temperature response curve. Observations relating to Panama tropical species were those presented by Slot and Winter (2017)[17]. The global gas-exchange data and that of Panama tropical species were accessed through the TRY database[65].

### Modelling of $CO_2$ assimilation

Net photosynthetic $CO_2$ fixation ($A_c$) was calculated using the FvCB $C_3$ photosynthesis model[4,29] using the equation:

$$A_c = \frac{(C - \Gamma^*)V_{cmax}}{C + K_c(1 + \frac{O}{K_o})} - R_L \qquad (2)$$

The $CO_2$ partial pressure at the site of fixation ($C$) was assumed to be an ambient atmospheric $CO_2$ partial pressure (40 Pa for current projections or as stated otherwise) multiplied by 0.7 to account for assumed intercellular $CO_2$ drawdown. We further assumed an infinite mesophyll conductance and no $CO_2$ conductance response to temperature. The partial pressure of oxygen in the atmosphere (O) was either set to ambient atmospheric partial pressure (21 kPa) or zero when assessing effects of photorespiration on $A_n$. Respiration in the light ($R_L$) was assumed to be 70% of dark respiration which was 1.29 μmol m$^{-2}$ s$^{-1}$ at 25 °C calculated from the relationship between nitrogen and respiration as presented in Atkin et al.[66], and its response to rising temperature calculated using a global quadratic model[67].

The $CO_2$ compensation point in the absence of $R_L$ ($\Gamma^*$; Pa), the Michaelis–Menten constant of Rubisco for $CO_2$ ($K_c$; Pa) and $O_2$ ($K_o$; kPa), and Rubisco $k_{cat}$ ($CO_2$ s$^{-1}$) at a given temperature were calculated using the Arrhenius equation:

$$Parameter = P_{25}e^{\left[\frac{E_a(T-25)}{R \cdot 298 \cdot T_K}\right]} \tag{3}$$

where $R$ is the gas constant (8.314 J K$^{-1}$ mol$^{-1}$), $T$ is the leaf temperature in degrees Celsius and $T_K$ the leaf temperature in Kelvin. $P_{25}$ is the parameter value at 25 °C and $E_a$ the activation energy in J mol$^{-1}$. The $P_{25}$ and $E_a$ values were based on multiple species both in vitro and in vivo presented in Galmés et al.[68] and Orr et al.[69] (Supplementary Table 1 and Supplementary Fig. 1). In regards to Orr et al.[69], cool was separated from warm growth kinetics by selecting species with a maximum temperature of the warmest quartile of below 25 °C or ≥ 25 °C, respectively. In regards to the Galmés et al.[68] dataset, cool was separated from warm growth kinetics by whether the maximum daily growth temperature was below 25 °C or ≥ 25 °C, respectively. Rubisco kinetics measured in vitro were converted from molar concentrations to partial pressures using Henry's law for solubilities and their temperature dependence:

$$Solubility = H^\circ e^{\left[-\triangle H\left(\frac{1}{T_K} - \frac{1}{298}\right)\right]} \tag{4}$$

Where $H°$ is the solubility at 25 °C (0.034 and 0.0013 mol L$^{-1}$ atm$^{-1}$ for $CO_2$ and $O_2$, respectively), $-\Delta H$ describes the temperature dependency of solubility (2400 and 1700 for $CO_2$ and $O_2$, respectively).

Based on the expected number of Rubisco catalytic sites ($n$), and the $k_{cat}$ of Rubisco at a given temperature determined above, we calculated $V_{cmax}$ as:

$$V_{cmax} = n \cdot k_{cat} \tag{5}$$

When models were fitted to individual species or combined species photosynthesis observations, the $n$ was obtained by solving the Eqs. 2 and 5 using parameters and the $A_n$ measured at 22 °C or closest observation above (temperatures where no Rubisco deactivation was observed):

$$n = \frac{(A_n + R_L)(C + K_c(1 + {}^O/K_o)}{k_{cat}(C - \Gamma^*)} \tag{6}$$

For a species independent estimate, $n$ was 26 µmol m$^{-2}$ based on 1.8 g m$^{-2}$ of leaf Rubisco content across a range of species[70] and a molecular weight of 70,000 g per active site[71]. To model the impact of RuBP regeneration-dependent assimilation ($A_r$)—the other commonly attributed limitation on $A_n$ at high temperatures[29]—we used the equation:

$$A_r = J\frac{C - \Gamma^*}{4C + 8\Gamma^*} - R_L \tag{7}$$

with the photosynthetic electron transport rate ($J$) and its temperature response modelled and parameterised using the equation of June et al.[47]:

$$J = J(T_o)e^{-\left(\frac{T-T_o}{\Omega}\right)^2} \tag{8}$$

Where $J(T_o)$ is the electron transport rate (µmol e m$^{-2}$ s$^{-1}$) at its temperature optimum ($T_o$), and $\Omega$ is the difference in temperature from $T_o$ at which $J$ declines to e$^{-1}$ (0.37)$J(T_o)$. $J(T_o)$, $T_o$ and $\Omega$ were the means of multiple species reported in June et al. (2004)[47] or derived from nonlinear least squares fits of Eq. 8 to other published temperature response curves of $J$ listed in Supplementary Table 3. Additionally, $T_o$ and $\Omega$ were supplemented with relativised temperature response curves of the quantum efficiency of PSII ($\phi$PSII). We considered the relative changes in $\phi$PSII to short-term changes in leaf temperature as equivalent to relative changes in $J$. Again, we separated the analysis into cool and warm grown plants based on mean day growth temperatures as outlined above. When fitting $A_r$ to individual species or combined species photosynthesis observations, we solved for $J(T_o)$ using Eqs. 7 and 8 using parameters and the $A_n$ measured at 22 °C or closest observation above:

$$J(T_o) = \frac{(A_n + R_L)(4C + 8\Gamma^*)}{e^{-\left(\frac{T-T_o}{\Omega}\right)^2}(C-\Gamma^*)} \tag{9}$$

$V_{cmax}$ based on $k_{cat}$ and Rubisco deactivation (Eq. 1) were compared with the peaked Arrhenius model of $V_{cmax}$ which is based on the temperature response of apparent $V_{cmax}$ derived from gas-exchange measurements and adjusted for growth temperature as described by Kattge and Knorr[23]:

$$V_{cmax} = V_{cmax}{}^{25}e^{\frac{H_a(T_K - T_{ref})}{T_{ref}RT_K}}\frac{1 + e^{\left(\frac{T_{ref}(\triangle S - H_d)}{T_{ref}R}\right)}}{1 + e^{\left(\frac{T_K\triangle S - H_d}{T_K R}\right)}} \tag{10}$$

Where $V_{cmax}{}^{25}$ was the value of $V_{cmax}$ at a reference temperature ($T_{ref}$) of 25 °C and set to 1. $H_d$ was the deactivation enthalpy and was set as 200 kJ mol$^{-1}$. The activation enthalpy ($H_a$) in J mol$^{-1}$ was calculated as 82,992-632×$T_{growth}$. The entropy factor ($\Delta S$) in J mol$^{-1}$ was calculated as 668.39-1.07×$T_{growth}$. Growth temperature ($T_{growth}$) was set to either 24 or 36 °C which provided the closest fit to $V_{cmax}$ derived from Eq. 1 for cool and warm grown plants.

All modelling and analysis were performed using R v.4.1.2 (2021-11-01) statistical software[72] with Rstudio graphical interface[73]. The R package readxl (v.1.3.1)[74] (https://CRAN.R-project.org/package=readxl) was used to analyse excel data files, and the package Metrics (v.0.1.4)[75] (https://CRAN.R-project.org/package=Metrics) was used to calculate root mean squared error (RMSE) and the bias between observed and predicted values.

### Reporting summary

Further information on research design is available in the Nature Portfolio Reporting Summary linked to this article.

## Data availability

The author and DOI information relating to the published articles from which all Rubisco kinetic, Rubisco activation state, net $CO_2$ assimilation, and electron transport rate temperature response curve data where analysed are presented in Supplementary Tables 1, 2, 3, and 4. The collated data are provided at https://doi.org/10.6084/m9.figshare.22661989 and with the model code (https://doi.org/10.5281/zenodo.7683559). The global individual $A_n$ vs leaf temperature[34] and Panama species dataset[17] are available from the TRY plant trait database http://www.try-db.org/TryWeb/Data.php under the dataset names "Global Leaf Gas Exchange Database (I)" and "Photosynthesis Temperature Response Panama".

## Code availability

A model script and $A_n$ data are accessible through GitHub (https://doi.org/10.5281/zenodo.7683559). The script provides model fits to $A_n$ temperature response curves using relevant model parameters based on whether the plant was grown below or above 25 °C. The script will generate graphs and provide indices of how well observed and model data match for each curve fit, comparable to the analysis presented in this study.

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

## Acknowledgements

This study was supported by an Australian Research Council DP22 grant (DP220101882 provided to A.P.S. and O.K.A.). We thank Assoc. Prof. Danielle Way for providing valuable feedback on the manuscript.

## Author contributions

A.P.S. conceived and drafted the initial manuscript. B.C.P., J.R.E., G.D.F., and O.K.A. provided substantive revisions and guidance in interpretation of the data.

## Competing interests

The authors declare no competing interests.
