## [Peer Review File · Nature Communications]

Reviewers' Comments:

Reviewer #1:

Remarks to the Author:

Scafaro et al incorporated Rubisco deactivation at moderately high temperatures into the FvCB model of photosynthesis to describe the temperature response of net CO₂ assimilation in C₃ species adapted to cool and warm environments. They used published data on the temperature response of photosynthesis and rubisco properties, including activation state and catalytic constants, for a variety of species. The model provides a good fit for experimental data obtained for most species evaluated, and a justification is given when the model does not fit the data, including thermophile species that may have a more thermal tolerant Rubisco activase and tropical species known to be subject to tighter stomatal control at elevated temperatures.

The key finding confirms previous research suggesting that Rubisco deactivation is a key limitation of photosynthesis at moderately high temperatures which plants are increasingly experiencing in the form of heat wave events. The manuscript adds evidence to this theory and is of value to the community by providing a framework to incorporate Rubisco deactivation into the FvCB model.

It would have been useful to include the loss of net CO₂ assimilation due to electron transport impairment in Fig 5, given the claim that Rubisco activation is more limiting than electron transport as the leaf temperature increases above optimum.

Some researchers would argue that Rca might be inactivated to prevent wasting ATP at moderately high temperatures, and/or that Rubisco inhibition may serve to protect the enzyme from proteolysis, as well as serve to maintain the balance between photosynthetic processes and protect PSII. These aspects warrant discussing here.

Similarly, it seems important to discuss the intersectionality between Rubisco activation and photorespiration. If a greater fraction of the Rubisco present in leaves remained active (through increasing the thermal tolerance of Rca), this would increase the rates of both photosynthesis and photorespiration. Would the impact of the latter become greater in this scenario?

The title could be more specific as the current version fails to highlight the novelty of the work, for example, something like: Rubisco deactivation is the key limitation of photosynthesis at moderately high temperatures in terrestrial C₃ plants.

The balance of text in the Results/Methods/Discussion could be better achieved. The start of the Results section reads as Methods, and supp fig 4 should probably be mentioned when talking about fig 3 in the Results, not just in the Discussion.

Additional suggestions/comments:

- Replace Rubisco active sites by Rubisco catalytic sites throughout and mention catalytic sites activated or deactivated, to avoid confusion between the site and its state.
- Replace reduction by decline. The difference is subtle, but decline seems more appropriate as reduction is frequently associated with an action while decline is a consequence of the environmental stress.
- Apply the same formatting criteria consistently throughout the manuscript. E.g., blue = cold, red = warm, etc. The use of red in Fig 2b is confusing – I'd use black lines here. The use of colour in Fig 3 to denote species is confusing, I'd use different symbols, etc.
- Use lower case k for kcat.
- Use T_{0.5} for the tempt at which the activity is halved (because h is typically associated with hour).
- Reference to supp fig 5 in the methods should be supp fig 4.
- Axis of supp fig 2 should have μmol instead of μmoles.

Reviewer #2:

Remarks to the Author:

This manuscript explores rubisco deactivation at high temperature as a predictor of photosynthetic sensitivity to temperature in global change scenarios. The authors conclude that reasonable expectations for how rubisco activation state will change in response to global warming fit observations better than other potential drivers, such as sensitivity of electron transport to elevated temperature. The authors have assembled a large amount of data from various sources

from which to make their claims. The conclusion they come to is that the increased photosynthetic rate expected by increasing CO₂ in the atmosphere will be significantly diminished or even eliminated once the deactivation process is accounted for on global models. I think the authors succeed in making their case.

I think that rubisco deactivation that is generally agreed to occur at high temperature can have consequences for predictions as provided here but also can be interpreted from a physiological perspective. If deactivation is acting as a circuit breaker when photorespiration rates are too high or as a way to prevent electron transport operating at its thermal maximum, then it is possible that genetic engineering or breeding for thermotolerant activase would be successful for plants growing in a high CO₂ world. This manuscript does not (and does not need to) address this concern although having one or two sentences about activase as a thermal fuse might be nice. The manuscript is aimed at describing an issue that needs to be considered in global modeling of climate change resulting from increased CO₂. In this regard the manuscript reaches the goal.

Minor issues

The phenomenon of CO₂ emissions in the light beyond those of photosynthesis are treated in a variety of ways. These should be made consistent. examples are line 47, 102, 247 "mitochondrial respiration", but old data from Calvin and new data from metabolic flux analysis shows that the TCA cycle is not the source of this CO₂. This is called RL (my preference) at lines 91 and 98. On line 223 this is called R_{light} (instead of RL).

Supplementary table 1 and Figure S1 The k of k_{cat} is capitalized. I think the convention of biochemists should be followed since these equations grow from biochemistry. k_{cat} (a rate constant) takes a lower case k to distinguish it from a Michaelis constant which is a ratio of lower case k's and takes an upper case K. (a lower case k is used in the main paper at lines 64, 77, 103, 114, 117,

Tom Sharkey

Reviewer #3:

Remarks to the Author:

This is useful addition to the modelling literature, but as claimed, it does not definitively show that Rubisco activase generally becomes limiting at elevated temperature, and that the modelled equation for an activase limitation fits in all but a few extremophile cases. It is also not the first paper to attempt to model limitations due to reductions in the Rubisco activation state. This has been done for over 30 years (and such literature is inadequately referenced – see for example Sage 1990 *Plant Physiol* 94:1728; Cen and Sage 1995, *Plant Physiol* 139: 979; Sage et al. *J Exp Bot* 59: 1581). The conclusions that "Rubisco activation explains the general reduction in photosynthesis above optimal leaf temperature" are over extrapolated and could be misleading for reasons stated below. As has been pointed out years ago, (and a point the authors acknowledge in their discussion of creosote bush responses), there are cases where reduction in activation state could limit A_n (by reducing V_c such that RuBP is saturating), and other cases where Rubisco deactivation is following other limitations (for example, RuBP regeneration rate is limiting and the reduction in activation state is a regulatory response to the decline in RuBP regeneration rate). It seems it would be more valuable to examine case by case situations where deactivation leads the limitation, and where it follows another limitations, and based on such an analysis, adjust the photosynthetic model to accommodate each situation. Fitting a metadata of the Rubisco activation state set to the Sharpe-Schoolfield equation does not get to the question of limitation, since we do not know the cause of the deactivation; it could be a lesion in activase that leads photosynthesis, it could be a regulatory reduction due to a limitation elsewhere, such that activase capacity follows the ultimate limitation, e.g. from an RuBP regeneration limitation.

Coming from such a prestigious group, the danger is the result could close the issue down and lead to a paradigm of general Rubisco activase limitation at high temperature that is premature and possibly incorrect. I recommend the authors develop their model into a more comprehensive analysis that explores more deeply possible limitations and regulatory feedbacks, and includes known parameters that contribute to limitation at high temperature, namely g_m and g_s. They might also try to model the RuBP regeneration limitation rate based on known liabilities at high temperature such as in the cytochrome b6f complex, and perhaps incorporate results from activase and cytochrome antisense studies, many of which were conducted at ANU. This may

require gathering novel empirical data to test the model predictions.

Some specifics:

1) The data spread above 30°C in Fig 1A is large, so lots of room for alternative limitations to be contributing. Summarizing this high spread with a fitted equation and then coming to a conclusion that the curve fit generally applies is premature it seems. This harkens to correlation exercises from phenomenological assessments that are overinterpreted to reflect causality, when any of a number of mechanisms could contribute.

2) The novel contribution is the fitting of a metadata set of activation state measurements from the literature with an exponential decay function developed by Schoolfield et al. 1981 and showing this can then predict response of photosynthesis to elevated temperature better than a modelled function for electron transport limited photosynthesis. Despite my concerns noted above, this is an important contribution and could be particularly valuable for predicting decay of A_n at elevated temperature. However, to be robust, and avoid the autocorrelation concern, the analysis should be conducted on the potential leading controls over A_n at elevated temperature, such as activase activity, cytochrome b6f stability, whole chain electron transport rate, or perhaps even enzymes in the Calvin cycle or TPU limitation should they be labile.

3. A key analysis is the electron transport rate limitation (W_j) uses June et al. 2004 for reversible temperature dependence. We are given little in how the June equation was applied to a diverse data set of cool and warm grown plants. How for example was J_{max} determined, and how was the critical omega termed derived? The omega term is the temp value where J is 0.37 of J_{max} . If Rubisco activase is limiting, and J is affected by this limitation, then the 0.37 value of omega would reflect the other limitation, conflating the assumption needed to accurately model the thermal response of J . Also, were all the species used in the J_{max} analysis the same as with the activation data? More details please so we can gauge the strength of the assessment.

4) Regarding the issue of regulation, activase itself is dependent upon electron transport rate and ATP synthesis, and so a decline in J at elevated temperature could lead to a decline in activase activity, which in turn causes deactivation. So the question ultimately is whether the reduction in rubisco activation state allows rubisco to increase its control over A_n , which in turn could reflect activase lability (and hence activase increases its control), or whether the control resides with electron transport and activase is simply following the leader. The analysis here cannot get to such an ultimate limitation, but there are a number of ways to get at this, such as using rca antisense plants, electron transport antisense plants, assess metabolite levels, and use various gas exchange methods such as in ref 8. Such measurements should be employed to test the robustness of the new algorithms.

I should also note that the majority of the early papers claiming activase limitations, such as in ref 9, ignored the possibility of electron transport limitations, and basically did a correlation analysis to show that activase lability is correlated with photosynthesis decline. However, as shown in Fig 5 of ref 6, if one models with an RuBP regeneration limitation included, the fit to Crafts-Bradner and Salvucci (2004) data is better than assuming just activase lability is limiting A_n .

5) A long-standing critique of the modelling approach is one can play with the model parameters until one gets a good fit, the so-called cherry picking approach (as noted in ref 8). In Cen and Sage, 2005, e.g. there is a bit of cherry-picking to show that electron transport could fit the A/T curves well. This may often be necessary, given certain ultimate limitations are simply not known, and other responses are just not right. For example, in Cen and Sage (2005) we found the Bernachi thermal responses of Rubisco V_{cmax} and J_{max} were inadequate for sweet potato, while a composite of spinach data for Rubisco provided a better fit. I wonder here if the parameterization of June et al. (2005) to make the case of a poor A_r fit (Fig 4) might be an example of cherry-picking to show consistency with the desired outcome. The lack of information on how eq 2 of June et al was applied does not allow me to overcome my concern.

It is also worth noting that Cen and Sage 2005 is listed as one of the data sources analyzed in this

study. Cen and Sage present decent evidence that Rubisco deactivation in sweet potato is not limiting but follows as a regulatory response to an electron transport limitation, based on direct in vitro assays of whole chain electron transport rate. The authors ignore this analysis and include the sugar beet data as a point in their conclusion that rubisco activation is generally limiting at high temperature, never mentioning the difference in conclusions. I disagree with this approach.

6) A concern I have with the early activase heat lability literature is a decline in activation state was noted at elevated temperature as activase lability was increasing, and this was presumed to cause the decline in A_n . But, as shown in Fig. 3A of Cen and Sage (2005) reducing CO_2 reactivates Rubisco, even at elevated temperature. If activase heat-lability were the cause of a decline in A_n , then its effect should be greatest at low CO_2 when the Rubisco control is strongest. Instead, low CO_2 , which causes Rubisco capacity to be limiting instead of RuBP regeneration, results in full activation. I point this out to demonstrate that there may not be a general activase limitation at elevated temperature. I agree there might be activase limitations, especially in non-acclimated species, but to conclude this is generally the case is unfounded, and runs the risk of shutting down enquiry into how evolution and acclimation adjust rubisco and activase to deal with high temperature liabilities. A more nuanced conclusion would be appropriate, one that points out the value of eq 6, but notes uncertainties that need to be examined.

5) Lines 44-47: The authors note CO_2 declines in leaves can affect A_n at elevated temperature, but then they chose to ignore temperature effects on g_m (assumed to be infinite, and no thermal response of conductance – line 95), which is inexplicable given we know temperature affects both g_m and g_s . For a paper trying to determine the rubisco deactivation (to include both decarbamylation and inhibitor binding) is the predominant limitation at elevated temperature, and then to ignore two known potential sources of limitation seems to be a major oversight (I actually had a thermal response paper rejected from PCE for assuming infinite g_m , about 10 years ago).

6) Certain aspects of the paper are not well developed, for example, the repeated reference to temperature effects on CO_2 delivery. The discussion of these points seemed scattered (e.g. see lines 317-330). Also, the discussion that activase is more limiting at elevated CO_2 on lines 330-340 did not make sense, appeared speculative and was poorly substantiated. This is based on modelled results in Fig 5b, but the model is not well explained. In any case, it is not very realistic to assume full activation of rubisco under conditions at elevated temperature where we know RuBP regeneration capacity has declined (as it generally does above the thermal optimum). This would be like arguing Rubisco activation state is limiting at low light and if rubisco were fully activated, the rate of A_n would be so much higher.

Also, lines 310-311 is speculative and adds little. Are they really claiming activase lability is responsible for the decline in electron transport above the thermal optimum, perhaps because activase interacts with PSI? If so, then a lot of excellent electron transport research would be invalid, including that in the June et al paper upon which a key part of the analysis is based.

7) Line 51-52 (“no analysis...”): This is a bit dismissive of the prior literature which have identified causes of A_n decline above the thermal optimum. They even cite some of the papers in support of an activase limitation at high temperature (refs 8 and 9), but then imply here that they don't really count as limitation studies. It is generally not a good strategy to build up the rationale by wholesale discounting prior work.

8) Line 58-64: This discussion is a bit convoluted and imprecise.

Why is this imprecise? A) V_{cmax} is the catalytic capacity of Rubisco at substrate saturation, as would be measured in vitro and usually expressed on a leaf area basis. It can be modelled (as the supporting refs did) under certain conditions (low CO_2 when Rubisco control is strong). k_{cat} is V_{cmax} divided by the number of functional active sites – so essentially the same parameter with a slight amendment. The authors seem to have forgotten the biochemical origin of V_{cmax} and instead equate it to a modelled parameter that reflects in vivo activity, which in the original model, is W_c . It has been argued (Sage 1990; Sage and Kubien 2007; Sage and Way 2008), that the V_{cmax} term can be amended by a deactivation function (as would occur during an activase limitation or regulatory reduction of the activation state) This would be described by V_{cmax}' , the

V_{cmax} that would correspond to a loss of functional catalytic sites in vivo. V_{cmax}' could then be modelled based on the assumption of that it is tracking a limitation in W_j, and the resulting modelled activation (V_{cmax}'/V_{cmax}) could be used to predict A. In turn, one could model V_{cmax}' as a function of any other limitation, such as activase, for example, by fitting the Schoolfield et al equation to activase activity above the thermal optimum. If activase were limiting, it would provide a better fit than if V_{cmax}' were tracking J. This is a way around the conundrum of modelling activation state responses without knowing whether activase ability or electron transport is determining the activation state. (Note, one could also make the same arguments at low light, where activation state declines, RuBP regeneration capacity declines, and activase capacity declines, seemingly in concert; but we know this would be false unless we have clear empirical evidence that activase is leading the limitation. The same standards should be held for supra-optimal thermal responses).

Second, the main source of the decline in activation state at elevated temperature is decarbamylation, not inhibitor binding. Inhibitors can get into the catalytic site and block catalysis, but this tends to be a reduction in K_{cat} in vitro, which is not observed. Inhibitors can act in concert with a decarbamylated catalytic site by slowing recarbamylation, and activase can help clean up the active sites so they can carbamylate. Reword to better reflect the deactivation mechanism.

9) The acclimation of activation state between cool and warm climate plants also reflects systems where electron transport rate is acclimating and/or adapting. It would seem that this needs to be effectively modelled as well, which is why the lack of explanation regarding the parameterization of June et al is even more problematic.

10)

11. The conclusion paragraph on lines 341 to 356 is a bit strong for the data. e.g. line 344-345 - tough to accept this conclusion when g_m and g_s are not modelled as a function of temperature. The use of "indicates" in line 347 reflects the suggestive nature of the results and contradicts the tone of certainty in the title of the paper and abstract. Line 349 - "drive mounting declines", is based on a flawed analysis as noted above. Line 350 - "speed of adaptation" is unsubstantiated, since this study did not address adaptive speed.

In sum, further model development and effective use of their model to predict clear examples where activase is known to show control would be a more robust approach. Perhaps avoid conclusions regarding activase being a general limitation until other limitations can be definitively ruled out, both experimentally and in theory. This is a long-standing expectation in photosynthetic limitation research that we all have had to adhere to lest we face unpleasant assessments by colleagues.

Sorry to be a downer, but I think you can have a much more powerful study if you build a more comprehensive model and evaluate it with some specific empirical tests of its predictions, as suggested above.

REVIEWER COMMENTS

Reviewer #1 (Remarks to the Author):

Scafaro et al incorporated Rubisco deactivation at moderately high temperatures into the FvCB model of photosynthesis to describe the temperature response of net CO₂ assimilation in C₃ species adapted to cool and warm environments. They used published data on the temperature response of photosynthesis and rubisco properties, including activation state and catalytic constants, for a variety of species. The model provides a good fit for experimental data obtained for most species evaluated, and a justification is given when the model does not fit the data, including thermophile species that may have a more thermal tolerant Rubisco activase and tropical species known to be subject to tighter stomatal control at elevated temperatures.

The key finding confirms previous research suggesting that Rubisco deactivation is a key limitation of photosynthesis at moderately high temperatures which plants are increasingly experiencing in the form of heat wave events. The manuscript adds evidence to this theory and is of value to the community by providing a framework to incorporate Rubisco deactivation into the FvCB model.

It would have been useful to include the loss of net CO₂ assimilation due to electron transport impairment in Fig 5, given the claim that Rubisco activation is more limiting than electron transport as the leaf temperature increases above optimum.

RESPONSE

This has been resolved as we have removed Fig. 5 and have added the comparison of electron transport with Rubisco deactivation throughout the revised manuscript.

Some researchers would argue that Rca might be inactivated to prevent wasting ATP at moderately high temperatures, and/or that Rubisco inhibition may serve to protect the enzyme from proteolysis, as well as serve to maintain the balance between photosynthetic processes and protect PSII. These aspects warrant discussing here.

RESPONSE

This has been addressed by the revisions which state the importance of electron transport (including the role of PSII) on co-limiting photosynthesis at high temperature. An example of where we highlight the importance of PSII is in the newly added section of the discussion (lines 309-323).

“The alignment of Rubisco deactivation and declines in J support a tight temperature dependent regulation between the two. Rca activity is modulated by ATP and inhibited by competitive binding of ADP^{57, 58, 59}. Declines in J with rising temperature due to electron transport imbalance, leakiness of thylakoid membranes, or other damage likely reduce stromal ATP concentrations. Lower stromal ATP concentrations reduce the active state of Rubisco⁶⁰, presumably through reduced Rca activity. Conversely, a lack of CO₂ fixation by Rubisco due to heat instability of Rca may lead to an accumulation of RuBP, reductant, and ATP. Recent analysis suggests that Cyt b_6/f tightly controls the dynamic flow of electrons between PSII and PSI, thus an accumulation of reductant and ATP would quickly downregulate Cyt b_6/f electron transfer⁵⁰. Interestingly, Rca has previously been found to associate with thylakoid membranes under heat stress in spinach (*Spinacia oleracea*, L.)⁶¹, and a recent report in rice noted a reduction in the quantum yield of photosystem I with overexpression of Rca⁶². Whether Rca and components of the electron transport chain interact directly during heat perturbation to coordinate downregulation of photosynthesis with rising temperature requires further exploration.”

Similarly, it seems important to discuss the intersectionality between Rubisco activation and photorespiration. If a greater fraction of the Rubisco present in leaves remained active (through increasing the thermal tolerance of Rca), this would increase the rates of both photosynthesis and photorespiration. Would the impact of the latter become greater in this scenario?

RESPONSE

We have addressed the relationship between Rubisco deactivation and photorespiration to make our interpretation more clearly understood. It is evident in Fig. 1c that photorespiration does increase to a greater extent with rising temperature. Although photorespiration increases with temperature, it does not impose a greater cost on assimilation than deactivation of Rubisco (or the decline in J). A_n which includes photorespiration but no deactivation of Rubisco or decline in J continues to increase to temperatures reaching above 45°C. In other words, an improved Rca would lead to greater photorespiration at higher temperatures but still an overall increase in photosynthesis.

We have made these points clearer in the Results and Discussion:

Lines 117-124: “Without a Rubisco deactivation term and in the absence of photorespiration (i.e. O₂ parameterised to zero), modelled A_c did not reach a T_{opt} below 50°C for either cool or warm grown plants (Fig. 1c). When accounting for photorespiration (i.e. O₂ parameterised to atmospheric concentrations of 21%)

but not Rubisco deactivation, A_c reached a T_{opt} at the relatively hot temperatures of 45.1 °C for cool and 46.6 °C for warm grown plants. With both photorespiration and Rubisco deactivation accounted for, the T_{opt} of A_c was 29.4 °C for cool and 32.7°C for warm grown plants and A_c declined sharply as temperatures exceeded these T_{opt} (Fig. 1c).”

Lines 267-276: “Rubisco specificity for CO₂ significantly shapes A_n under moderate, sustained warming, and photorespiration becomes a greater contributor to CO₂ loss as leaf temperature rises (Fig. 1c). However, photorespiration cannot account for the extent of A_n decline that occurs as leaf temperature exceeds T_{opt} (Fig. 1c). Indeed, model predictions of A_n that account for photorespiration but not for Rubisco deactivation or J limitations far exceed observed A_n , estimating a T_{opt} of 45°C in cool grown plants (Fig. 1c). It is therefore unlikely that Rubisco deactivation and J are downregulated as a mechanism to limit photorespiratory CO₂ loss considering both contribute far more to declines in A_n above T_{opt} than photorespiration does.”

The title could be more specific as the current version fails to highlight the novelty of the work, for example, something like: Rubisco deactivation is the key limitation of photosynthesis at moderately high temperatures in terrestrial C3 plants.

RESPONSE

Changed title to be more specific by including the text terrestrial C3 plants but have not used the suggested title due to the change in interpretation of the results away from suggesting Rubisco deactivation is the rate limiting step.

The balance of text in the Results/Methods/Discussion could be better achieved. The start of the Results section reads as Methods, and supp fig 4 should probably be mentioned when talking about fig 3 in the Results, not just in the Discussion.

RESPONSE

We wanted to capture the way we modelled Rubisco deactivation in the results as it is central to the findings presented. But this has led to the start of the results being heavy with background methodology relating to all the equations used and their parameterisation. To find a better balance, we have only kept the central V_{cmax} calculation based on the Sharpe-Schoolfield equation for enzyme deactivation in the results and moved all the other equations and their parameterisation to the methods. We have now mentioned Fig. S4 in results in a more appropriate position.

Additional suggestions/comments:

- Replace Rubisco active sites by Rubisco catalytic sites throughout and mention catalytic sites activated or deactivated, to avoid confusion between the site and its state.

RESPONSE

Changed as suggested

- Replace reduction by decline. The difference is subtle, but decline seems more appropriate as reduction is frequently associated with an action while decline is a consequence of the environmental stress.

RESPONSE

Changed 'reduction' to 'decline' throughout text

- Apply the same formatting criteria consistently throughout the manuscript. E.g., blue = cold, red = warm, etc. The use of red in Fig 2b is confusing – I'd use black lines here. The use of colour in Fig 3 to denote species is confusing, I'd use different symbols, etc.

RESPONSE

Changed as suggested. We have kept colours to differentiate plant functional types due to the substantial amount of overlap in points but have different symbols as well. The colours used for plant functional types do not include blue or red to limit confusion with the cool and warm grown model fits.

- Use lower case k for k_{cat} .

RESPONSE

Checked and corrected any upper case K_{cat} .

- Use $T_{0.5}$ for the temp at which the activity is halved (because h is typically associated with hour).

RESPONSE

Changed as suggested

- Reference to supp fig 5 in the methods should be supp fig 4.

RESPONSE

Fixed

- Axis of supp fig 2 should have μmol instead of μmoles .

RESPONSE

Fixed

Reviewer #2 (Remarks to the Author):

This manuscript explores rubisco deactivation at high temperature as a predictor of photosynthetic sensitivity to temperature in global change scenarios. The authors conclude that reasonable expectations for how rubisco activation state will change in response to global warming fit observations better than other potential drivers, such as sensitivity of electron transport to elevated temperature. The authors have assembled a large amount of data from various sources from which to make their claims. The conclusion they come to is that the increased photosynthetic rate expected by increasing CO₂ in the atmosphere will be significantly diminished or even eliminated once the deactivation process is accounted for on global models. I think the authors succeed in making their case. I think that rubisco deactivation that is generally agreed to occur at high temperature can have consequences for predictions as provided here but also can be interpreted from a physiological perspective. If deactivation is acting as a circuit breaker when photorespiration rates are too high or as a way to prevent electron transport operating at its thermal maximum, then it is possible that genetic engineering or breeding for thermotolerant activase would be successful for plants growing in a high CO₂ world. This manuscript does not (and does not need to) address this concern although having one or two sentences about activase as a thermal fuse might be nice.

RESPONSE

As with reviewer 1 and 3, the interaction between activation state as a fuse to protect against unwanted photorespiration and overreduction due to excessive electron transport is clarified in Fig. 1. We now argue that Rubisco deactivation and declines in J are coordinated and co-limiting. In terms of photorespiration, as mentioned in regards to Reviewer 1, it seems unlikely to us that Rubisco deactivation is a mechanism to reduce photorespiration considering that photorespiration imposes less of a cost to assimilation than what deactivation and J do. This is clarified in the revised Fig. 1. We further clarify this point in the Discussion in Lines 267-276 as stated above.

The manuscript is aimed at describing an issue that needs to be considered in global modeling of climate change resulting from increased CO₂. In this regard the

manuscript reaches the goal.

Minor issues

The phenomenon of CO₂ emissions in the light beyond those of photosynthesis are treated in a variety of ways. These should be made consistent. examples are line 47, 102, 247 “mitochondrial respiration”, but old data from Calvin and new data from metabolic flux analysis shows that the TCA cycle is not the source of this CO₂.

This is called RL (my preference) at lines 91 and 98. On line 223 this is called R_{light} (instead of RL).

RESPONSE

Change all mention of mitochondrial respiration to respiration in the light and abbreviated to R_L .

Supplementary table 1 and Figure S1 The k of k_{cat} is capitalized. I think the convention of biochemists should be followed since these equations grow from biochemistry. k_{cat} (a rate constant) takes a lower case k to distinguish it from a Michaelis constant which is a ratio of lower case k's and takes an upper case K. (a lower case k is used in the main paper at lines 64, 77, 103, 114, 117, Tom Sharkey

RESPONSE

Checked and corrected.

Reviewer #3 (Remarks to the Author):

This is useful addition to the modelling literature, but as claimed, it does not definitively show that Rubisco activase generally becomes limiting at elevated temperature, and that the modelled equation for an activase limitation fits in all but a few extremophile cases. It is also not the first paper to attempt to model limitations due to reductions in the Rubisco activation state. This has been done for over 30 years (and such literature is inadequately referenced – see for example Sage 1990 Plant Physiol 94:1728; Cen and Sage 1995, Plant Physiol 139: 979; Sage et al. J Exp Bot 59: 1581).

RESPONSE

We realise that some important work that covers similar aspects of predicting photosynthesis limitations based on Rubisco deactivation and electron transport

were overlooked. We have now referred to this literature, including the papers that are specifically mentioned above.

Lines 42-49: “Further experiments and modelling have identified A_c as the rate limiting step in some instances, while other studies have implicated A_r due to declines in J with rising temperatures^{6, 11, 12}. Whether A_c or A_r determines A_n above the T_{opt} is often attributed to interspecific differences or environmental factors such as nitrogen availability, growth temperature, and ambient CO_2 concentration^{6, 12, 13}. An alternative possibility is that A_c and A_r are both regulated to be co-limiting. For example, Sage (1990)¹⁴ proposed and observed¹⁵ synchronised A_c and A_r biochemical adjustments within minutes of altering irradiation and CO_2 concentrations.”

Lines 81-83: “Alternatively, a more satisfying reconciliation of this inconsistency is to calculate V_{cmax} based on the k_{cat} of Rubisco and its deactivation based on biochemical observations^{14, 30}.”

The conclusions that “Rubisco activation explains the general reduction in photosynthesis above optimal leaf temperature” are over extrapolated and could be misleading for reasons stated below. As has been pointed out years ago, (and a point the authors acknowledge in their discussion of creosote bush responses), there are cases where reduction in activation state could limit A_n (by reducing V_c such that RuBP is saturating), and other cases where Rubisco deactivation is following other limitations (for example, RuBP regeneration rate is limiting and the reduction in activation state is a regulatory response to the decline in RuBP regeneration rate).

It seems it would be more valuable to examine case by case situations where deactivation leads the limitation, and where it follows another limitations, and based on such an analysis, adjust the photosynthetic model to accommodate each situation.

RESPONSE

We have reorientated the manuscript away from claims of what is the limiting step that sets assimilation and rather explored the tight coordination between deactivation and electron transport limitations on assimilation across a global dataset.

Fitting a metadata of the Rubisco activation state set to the Sharpe-Schoolfield equation does not get to the question of limitation, since we do not know the

cause of the deactivation; it could be a lesion in activase that leads photosynthesis, it could be a regulatory reduction due to a limitation elsewhere, such that activase capacity follows the ultimate limitation, e.g. from an RuBP regeneration limitation.

RESPONSE

In the revised manuscript, we explore the idea that Rubisco deactivation and electron transport are tightly controlled, and both set limits on assimilation. This includes a new paragraph in the discussion that explores ways in which one will impact the other leading to a similar response for both to temperature.

Lines 309-323: “The alignment of Rubisco deactivation and declines in J support a tight temperature dependent regulation between the two. Rca activity is modulated by ATP and inhibited by competitive binding of ADP^{57, 58, 59}. Declines in J with rising temperature due to electron transport imbalance, leakiness of thylakoid membranes, or other damage likely reduce stromal ATP concentrations. Lower stromal ATP concentrations reduce the active state of Rubisco⁶⁰, presumably through reduced Rca activity. Conversely, a lack of CO₂ fixation by Rubisco due to heat instability of Rca may lead to an accumulation of RuBP, reductant, and ATP. Recent analysis suggests that Cyt b_6/f tightly controls the dynamic flow of electrons between PSII and PSI, thus an accumulation of reductant and ATP would quickly downregulate Cyt b_6/f electron transfer⁵⁰. Interestingly, Rca has previously been found to associate with thylakoid membranes under heat stress in spinach (*Spinacia oleracea*, L.)⁶¹, and a recent report in rice noted a reduction in the quantum yield of photosystem I with overexpression of Rca⁶². Whether Rca and components of the electron transport chain interact directly during heat perturbation to coordinate downregulation of photosynthesis with rising temperature requires further exploration.”

Coming from such a prestigious group, the danger is the result could close the issue down and lead to a paradigm of general Rubisco activase limitation at high temperature that is premature and possibly incorrect. I recommend the authors develop their model into a more comprehensive analysis that explores more deeply possible limitations and regulatory feedbacks, and includes known parameters that contribute to limitation at high temperature, namely g_m and g_s .

RESPONSE

One of the most striking aspects of the model that we presented is that mesophyll and stomatal limitations (g_m and g_s , respectively) are not required to explain a decline in assimilation with heating across multiple species. We acknowledge that g_m and g_s may contribute and even be the main driver of loss of A_n above a T_{opt} ,

but our central point is that it is not a necessity to explain widely observed declines in photosynthesis with heat. We have reinforced this point in multiple sections in the Introduction and Discussion to make it clear that we do not claim that in all circumstances Rubisco deactivation or electron transport is the driver of declines in A_n , only that they will be in the absence of these other limitations. For example:

Lines 50=56: “Not only can the capacity of Rubisco to fix CO_2 and light dependent generation of RuBP be impaired by heat, but the availability of CO_2 substrate at the site of assimilation can fall and become limiting. Reduced A_n due to falling intercellular and chloroplast CO_2 concentrations following heat-associated rises in vapor pressure differences between leaves and air have been observed^{16, 17, 18}. Additionally, foliar CO_2 loss from photorespiration and respiration in the light (R_L) may contribute substantially to declining A_n under high temperature¹⁹, as both processes rise sharply with warming^{20, 21}.”

Lines 252-264: “Although we removed the confounding influence of water stress in this study, water availability remains influential in reducing A_n in species that close their stomata to preserve water, particularly during hot and dry conditions when vapour pressure deficit is high and soil moisture is low^{16, 39}. Indeed, our amended A_c model overestimated the temperature at which A_n began to decline when compared to tropical tree and lianas species in Panama (Supplementary Fig. 5). This was consistent with the published declines in stomatal conductance and intercellular CO_2 concentrations in response to leaf heating for these same species¹⁷. Another aspect of CO_2 conductance that can influence A_n is the rate of CO_2 diffusion from intercellular airspaces to the site of chloroplasts, termed mesophyll conductance. Mesophyll conductance appears to either increase or remain constant with short-term rises in leaf temperature across many species⁴¹. This will contribute to intercellular CO_2 drawdown and may exacerbate Rubisco CO_2 substrate limitations when water is limited and temperatures hot.”

Lines 349-353: “Finally, we demonstrated that neither CO_2 substrate supply limitation nor photorespiratory CO_2 loss was needed to explain high temperature-induced decreases in A_n . However, many future heatwaves are likely to coincide with drought, and drought will reduce CO_2 conductance and increase photorespiratory CO_2 loss, exacerbating the stress caused by Rubisco deactivation and declines in J .”

They might also try to model the RuBP regeneration limitation rate based on known liabilities at high temperature such as in the cytochrome b6f complex, and perhaps incorporate results from activase and cytochrome antisense studies,

many of which were conducted at ANU. This may require gathering novel empirical data to test the model predictions.

RESPONSE

In our experience and in view of the literature, it is difficult to empirically determine the point at which electron transport and deactivation set a limit on A_n as both are so highly coordinated. Instead, our reemphases on the predictive power of Rubisco activation state but the likelihood that it is in coordination with electron transport limitations makes the need for teasing apart the limitation not essential to the main findings of the manuscript.

Some specifics:

1) The data spread above 30°C in Fig 1A is large, so lots of room for alternative limitations to be contributing. Summarizing this high spread with a fitted equation and then coming to a conclusion that the curve fit generally applies is premature it seems. This harkens to correlation exercises from phenomenological assessments that are overinterpreted to reflect causality, when any of a number of mechanisms could contribute.

RESPONSE

Spread is likely due to difficulty in conducting biochemical analysis of Rubisco activation state and the fact that it is a composite of many species that have been grown at a range of temperatures. It was not possible to separate by tighter growth ranges because of the limited data, but we suggest that this would have reduced the spread. Our compromise was to fit a model to the broader categories of below and above 25°C but future studies should refine this as we agree that the fits can be made tighter. We have added this point to the manuscript:

Lines 343-349: “Although T_{opt} is known to shift with growth temperature at a finer scale than simply below or above 25°C, the limited number of published Rubisco activation state and J temperature curves prohibits model parameterisation that would allow predictions of finer scale adjustments in A_n to changing growth temperature. Further studies that characterise the temperature dependence of Rubisco deactivation and temperature dependence of J – ideally from a wide spread of plant functional types – will improve the accuracy of the models we present.”

Our revisions that accept there was no significant difference in predictability between deactivation and electron transport remove the importance attributed to the slight difference in predictions between deactivation and electron transport. In the revised manuscript, we tested this statistically; the analysis

revealed no significant difference between the two models, confirming an overinterpretation of the Rubisco activation state data (refer to the new Fig. 3).

2) The novel contribution is the fitting of a metadata set of activation state measurements from the literature with an exponential decay function developed by Schoolfield et al. 1981 and showing this can then predict response of photosynthesis to elevated temperature better than a modelled function for electron transport limited photosynthesis. Despite my concerns noted above, this is an important contribution and could be particularly valuable for predicting decay of A_n at elevated temperature. However, to be robust, and avoid the autocorrelation concern, the analysis should be conducted on the potential leading controls over A_n at elevated temperature, such as activase activity, cytochrome b_6/f stability, whole chain electron transport rate, or perhaps even enzymes in the Calvin cycle or TPU limitation should they be labile.

RESPONSE

We have added a more comprehensive analysis of the electron transport data and its role in photosynthesis limitations at high temperature. This includes adding and reporting more temperature response curves of J to improve our model power. It includes a statistical comparison of predictability between the A_c and A_r models (Fig. 3). The outcome is that we do not overemphasise the limitation caused by deactivation and we explore to a greater extent the limitations that could be imposed on thylakoid reactions:

Lines 287-306: "The dynamic and reversible decline in J at high temperatures⁴⁶ has been linked to heat susceptibility of thylakoid membranes and their constituents^{7, 11, 47}. The oxygen evolving complex of PSII and the cytochrome b_6/f complex (Cyt b_6/f) seem particularly important in setting dynamic temperature-effected rates of J ^{48, 49, 50}. There are four Mn atoms per PSII reaction centre responsible for oxidation of H_2O . Mn is held by 33 kDa D1 proteins. Prolonged heat stress can dislodge D1 proteins and subsequently Mn^{2+} ions from the oxygen evolving complex of higher plants, resulting in a decline in J ^{51, 52, 53}. The disruption in electron accepting ability of PSII leads to the reaction centre being overly oxidised (P_{680}^+) and conducive to ROS formation, which can impair D1 protein synthesis and further diminishes PSII functionality⁵⁴. The heat sensitivity of the PSII oxygen evolving complex makes it a key reason for why J declines under high leaf temperatures. Heat damage to thylakoid membranes is not confined to PSII. Moderate temperature of 40°C is shown to disrupt the balance of electron flow between PSII and PSI which is controlled by Cyt b_6/f , and through damage or regulation, the flow of electrons through Cyt b_6/f leads to an overreduction of PSI upon heat exposure⁵⁵. With an overly reduced PSI, cyclic electron flow is upregulated as a means of dissipating electrons and preventing irreversibly

damage to the stroma^{7, 55, 56}. However, cyclic electron flow is insufficient to maintain A_n in the absence of linear electron flow since it does not produce the necessary NADPH to run the Calvin-Benson cycle⁵⁶. This is a strong indicator that the imbalance in electron flow as temperatures exceed the T_{opt} contributes to declining J and subsequently A_r .”

3. A key analysis is the electron transport rate limitation (W_j) uses June et al. 2004 for reversible temperature dependence. We are given little in how the June equation was applied to a diverse data set of cool and warm grown plants. How for example was J_{max} determined, and how was the critical omega termed derived? The omega term is the temp value where J is 0.37 of J_{max} . If Rubisco activase is limiting, and J is affected by this limitation, then the 0.37 value of omega would reflect the other limitation, conflating the assumption needed to accurately model the thermal response of J . Also, were all the species used in the J_{max} analysis the same as with the activation data? More details please so we can gauge the strength of the assessment.

RESPONSE

We have clarified in the methods (Equations 8 and 9) and in a new Supplementary Table 3 the references and metadata relating to the J analysis.

4) Regarding the issue of regulation, activase itself is dependent upon electron transport rate and ATP synthesis, and so a decline in J at elevated temperature could lead to a decline in activase activity, which in turn causes deactivation. So the question ultimately is whether the reduction in rubisco activation state allows rubisco to increase its control over A_n , which in turn could reflect activase lability (and hence activase increases its control), or whether the control resides with electron transport and activase is simply following the leader. The analysis here cannot get to such an ultimate limitation, but there are a number of ways to get at this, such as using rca antisense plants, electron transport antisense plants, assess metabolite levels, and use various gas exchange methods such as in ref 8. Such measurements should be employed to test the robustness of the new algorithms.

RESPONSE

Great point about ATP and which loss of function lead the other. With our expanded analysis of J and our modified conclusions that point to both being equal predictors of photosynthesis decline we have also refined our speculation on how the two may be controlled, including through ATP regulation as provided in previous responses (lines 307:321 of text).

I should also note that the majority of the early papers claiming activase limitations, such as in ref 9, ignored the possibility of electron transport limitations, and basically did a correlation analysis to show that activase lability is correlated with photosynthesis decline. However, as shown in Fig 5 of ref 6, if one models with an RuBP regeneration limitation included, the fit to Crafts-Bradner and Salvucci (2004) data is better than assuming just activase lability is limiting A_n .

RESPONSE

We take the point that depending on the analysis A_n limitations have been attributed to both activation and electron transport limitations. This aligns with our revised analysis where the truth is likely to be that both set limits.

5) A long-standing critique of the modelling approach is one can play with the model parameters until one gets a good fit, the so-called cherry picking approach (as noted in ref 8). In Cen and Sage, 2005, e.g. there is a bit of cherry-picking to show that electron transport could fit the A/T curves well. This may often be necessary, given certain ultimate limitations are simply not known, and other responses are just not right. For example, in Cen and Sage (2005) we found the Bernachi thermal responses of Rubisco V_{cmax} and J_{max} were inadequate for sweet potato, while a composite of spinach data for Rubisco provided a better fit. I wonder here if the parameterization of June et al. (2005) to make the case of a poor A_r fit (Fig 4) might be an example of cherry-picking to show consistency with the desired outcome. The lack of information on how eq 2 of June et al was applied does not allow me to overcome my concern.

RESPONSE

This could be true. The lead author has a background in Rubisco activase research and thus was overly focused on that aspect of the analysis. We have now rectified this by adding data and analysis relating to electron transport. This includes providing the June et al. (2004) model (methods) and adding extra temperature response curves of J from the literature (Supplementary Table 3). It includes adding RuBP regenerated limited assimilation predictions of all model fits to observations (Figs. 1,2,4) and statistically comparing the predictions based on A_c and A_r (Fig. 3).

It is also worth noting that Cen and Sage 2005 is listed as one of the data sources analyzed in this study. Cen and Sage present decent evidence that Rubisco

deactivation in sweet potato is not limiting but follows as a regulatory response to an electron transport limitation, based on direct in vitro assays of whole chain electron transport rate. The authors ignore this analysis and include the sugar beet data as a point in their conclusion that rubisco activation is generally limiting at high temperature, never mentioning the difference in conclusions. I disagree with this approach.

RESPONSE

We thank the reviewer for these comments. However, we are a bit unsure of what is meant by the comment, as we did not directly cite Cen and Sage as being in support of our conclusions. Rather, we simply extracted the photosynthesis temperature response of sweet potato, along with similar data from multiple papers and species. We used this data as a tool to fit our model to observations. This was to build a widely applicable model with observations from a range of species. We did not factor in any conclusions from the papers we were extracting the observations from as to not bias the data based on previous conclusions. Thus, we do not feel that changes are necessary to address the reviewer's concern.

We disagree with the statement "Cen and Sage present decent evidence that Rubisco deactivation in sweet potato is not limiting but follows as a regulatory response to an electron transport limitation, based on direct in vitro assays of whole chain electron transport rate."

Busch and Sage (2017) ([doi: 10.1111/nph.14258](https://doi.org/10.1111/nph.14258)) repeat the experiment on sweet potato using different methodology and conclude that Rubisco deactivation is the main limitation at warmer temperatures. They state that the difference in conclusions between the two studies is likely due to differences in methodology and not due to biology:

From Busch and Sage (2017) : "*Cen & Sage (2005) attributed a deactivation of Rubisco observed at high temperatures to a regulatory feedback on Rubisco from limitations in TPU and RuBP regeneration capacity. Because our sweet potato plants and growth conditions were identical to those used in their study, we conclude that the differences between the respective predicted limitations are not biological, but reflect different analytical approaches.*"

We do refer to these two papers in the introduction in the context of the uncertainty that exists in the literature about what is the limiting step of assimilation at warmer temperatures:

Lines 42-46: “Further experiments and modelling have identified A_c as the rate limiting step in some instances, while other studies have implicated A_r due to declines in J with rising temperatures^{6, 8, 11, 12.}”

⁸ Busch and Sage (2017)

¹¹ Cen and Sage (2005)

6) A concern I have with the early activase heat lability literature is a decline in activation state was noted at elevated temperature as activase lability was increasing, and this was presumed to cause the decline in A . But, as shown in Fig. 3A of Cen and Sage (2005) reducing CO_2 reactivates Rubisco, even at elevated temperature. If activase heat-lability were the cause of a decline in A , then its effect should be greatest at low CO_2 when the Rubisco control is strongest. Instead, low CO_2 , which causes Rubisco capacity to be limiting instead of RuBP regeneration, results in full activation. I point this out to demonstrate that there may not be a general activase limitation at elevated temperature. I agree there might be activase limitations, especially in non-acclimated species, but to conclude this is generally the case is unfounded, and runs the risk of shutting down enquiry into how evolution and acclimation adjust rubisco and activase to deal with high temperature liabilities. A more nuanced conclusion would be appropriate, one that points out the value of eq 6, but notes uncertainties that need to be examined.

RESPONSE

We agree that the limitation proposed to be Rubisco activation alone is an overstatement. As mentioned with our other responses and evident in the revisions, we no longer claim that Rubisco deactivation alone is the limiting factor in nature.

5) Lines 44-47: The authors note CO_2 declines in leaves can affect A_n at elevated temperature, but then they chose to ignore temperature effects on g_m (assumed to be infinite, and no thermal response of conductance – line 95), which is inexplicable given we know temperature affects both g_m and g_s . For a paper trying to determine the rubisco deactivation (to include both decarbamylation and inhibitor binding) is the predominant limitation at elevated temperature, and then to ignore two known potential sources of limitation seems to be a major oversight (I actually had a thermal response paper rejected from PCE for assuming infinite g_m , about 10 years ago).

RESPONSE

The point we make is that by assuming a constant supply of CO₂ at the site of chloroplast in our model, we are demonstrating that the decline in A_n above T_{opt} can be attributed to biochemical limitations without the need for CO₂ conductance limitations (which would include both g_s and g_m). This is becoming apparent in many recent studies (please refer to the articles cited: 34,37,38,39,40). It is not to say g_m and g_s are never responsible for limiting A_n . Please refer to the comment above where we provide sections of the text demonstrating that loss of photosynthesis at high temperature can be linked to reduced CO₂ supply (e.g. Supplementary Fig. 5). To give more background on the specific role that g_m may play in temperature limitations we have added to the Discussion:

Lines261-264: “Another aspect of CO₂ conductance that can influence A_n is the rate of CO₂ diffusion from intercellular airspaces to the site of chloroplasts, termed mesophyll conductance. Mesophyll conductance seems to either increase or remain constant with short-term rises in leaf temperature across many species⁴¹. This will contribute to intercellular CO₂ drawdown and may exacerbate Rubisco CO₂ substrate limitations when water is limited and temperatures hot.”

6) Certain aspects of the paper are not well developed, for example, the repeated reference to temperature effects on CO₂ delivery. The discussion of these points seemed scattered (e.g. see lines 317-330). Also, the discussion that activase is more limiting at elevated CO₂ on lines 330-340 did not make sense, appeared speculative and was poorly substantiated. This is based on modelled results in Fig 5b, but the model is not well explained. In any case, it is not very realistic to assume full activation of rubisco under conditions at elevated temperature where we know RuBP regeneration capacity has declined (as it generally does above the thermal optimum). This would be like arguing Rubisco activation state is limiting at low light and if rubisco were fully activated, the rate of A_n would be so much higher.

RESPONSE

We have removed the analysis relating to future CO₂ concentrations and growth temperatures (previous Fig. 5). This analysis was more speculative in that it assumed Rubisco deactivation would hold with changes in CO₂ concentrations. We understand the point that at higher CO₂ concentrations RuBP regeneration may become a greater influence on A_n .

Also, lines 310-311 is speculative and adds little. Are they really claiming activase lability is responsible for the decline in electron transport above the thermal

optimum, perhaps because activase interacts with PSI? If so, then a lot of excellent electron transport research would be invalid, including that in the June et al paper upon which a key part of the analysis is based.

RESPONSE

We have modified this section to be less speculative but have not removed the citations or general statement that Rubisco activase (Rca) may in some way interact with parts of electron transport. The papers cited demonstrate an association between Rca and the photosystems and are peer reviewed so we do not dismiss them as unfounded. It adds support to our analysis that Rubisco activation and electron transport are highly synchronised. We are not suggesting that Rca is responsible for electron transport rates, only that under the specific conditions of heat damage, Rca may be one contributing part, amongst others, that regulates or is regulated by electron transport. We raise this as a postulation and do not state it emphatically:

Lines 316-321: “Interestingly, Rca has previously been found to associate with thylakoid membranes under heat stress in spinach (*Spinacia oleracea*, L.)⁵⁶, and a recent report in rice noted a reduction in the quantum yield of photosystem I with overexpression of Rca⁵⁷. Whether Rca and components of the electron transport chain interact directly during heat perturbation to coordinate downregulation of photosynthesis with rising temperature requires further exploration.”

7) Line 51-52 (“no analysis....”): This is a bit dismissive of the prior literature which have identified causes of A_n decline above the thermal optimum. They even cite some of the papers in support of an activase limitation at high temperature (refs 8 and 9), but then imply here that they don't really count as limitation studies. It is generally not a good strategy to build up the rationale by wholesale discounting prior work.

RESPONSE

The statements have been modified to be less emphatic. This text was meant to convey that it is difficult to pinpoint a more generalised limitation on photosynthesis because of the multitude of potential reasons for A_n to decline. It was not meant to suggest no accurate accounts of why photosynthesis declines with temperature have been made before. We have modified the sentence to read:

Lines 58-60: “Despite, or perhaps because of, the numerous aspects of photosynthetic metabolism that are impaired by heat, it is difficult to establish a

general predictor for the decline in A_n at relatively moderate temperature applicable across many higher plants.”

8) Line 58-64: This discussion is a bit convoluted and imprecise.

Why is this imprecise? A) V_{cmax} is the catalytic capacity of Rubisco at substrate saturation, as would be measured in vitro and usually expressed on a leaf area basis. It can be modelled (as the supporting refs did) under certain conditions (low CO₂ when Rubisco control is strong).

RESPONSE

Yes, we agree. But V_{cmax} is ubiquitously used in the literature to refer to the *in vivo* estimation of V_{cmax} . We were using it in the same context. Our manuscript highlights that they are not one and the same. We have modified the language to make this clearer throughout the text including in the example given:

Lines 62-65: “**Gas-exchange estimates** of V_{cmax} increase exponentially with temperature before peaking and then declining at higher temperatures; the point of decline is influenced by acclimation to growth temperature^{23, 24}. This decline in **apparent** V_{cmax} is not explained by susceptibility of Rubisco to high temperature.

K_{cat} is V_{cmax} divided by the number of functional active sites – so essentially the same parameter with a slight amendment. The authors seem to have forgotten the biochemical origin of V_{cmax} and instead equate it to a modelled parameter that reflects in vivo activity, which in the original model, is W_c .

We have not forgotten the biochemical origin of V_{cmax} as the entire analysis presented in the manuscript is essentially using the biochemical origin of V_{cmax} (please refer to Equations 1 and 5.)

It has been argued (Sage 1990; Sage and Kubien 2007; Sage and Way 2008), that the V_{cmax} term can be amended by a deactivation function (as would occur during an activase limitation or regulatory reduction of the activation state) This would be described by V_{cmax}' , the V_{cmax} that would correspond to a loss of functional catalytic sites in vivo. V_{cmax}' could then be modelled based on the assumption of that it is tracking a limitation in W_j , and the resulting modelled activation (V_{cmax}'/V_{cmax}) could be used to predict A . In turn, one could model V_{cmax}' as a function of any other limitation, such as activase, for example, by fitting the Schoolfield et al equation to activase activity above the thermal optimum. If activase were limiting, it would provide a better fit than if V_{cmax}' were tracking J . This is a way around the conundrum of modelling activation state responses

without knowing whether activase ability or electron transport is determining the activation state. (Note, one could also make the same arguments at low light, where activation state declines, RuBP regeneration capacity declines, and activase capacity declines, seemingly in concert; but we know this would be false unless we have clear empirical evidence that activase is leading the limitation. The same standards should be held for supra-optimal thermal responses).

RESPONSE

Thank you for reminding us of these related studies. We agree that they are similar in that they account for Rubisco deactivation and model the temperature response of A_n at individual species levels through means of accounting for biochemically observed declines in Rubisco activation state. We have now cited these articles as previous examples of modelling activation state impacts on the response of photosynthesis to environmental perturbations:

Lines 42-44: "Further experiments and modelling have in some instances identified A_c as the rate limiting step, and in other studies A_r due to declines in J with rising temperatures^{6, 11, 12}."

Lines 46-49: "An alternative possibility is that both A_c or A_r are co-limiting and regulated to be so. For example, Sage (1990)¹⁴ proposed and observed¹⁵ synchronised A_c and A_r biochemical adjustments within minutes of altering irradiation and CO₂ concentrations."

Lines 81-83: "Alternatively, a more satisfying reconciliation of this inconsistency is to calculate V_{cmax} based on the k_{cat} of Rubisco and its deactivation based on biochemical observations^{14, 30}."

Second, the main source of the decline in activation state at elevated temperature is decarbamylation, not inhibitor binding. Inhibitors can get into the catalytic site and block catalysis, but this tends to be a reduction in K_{cat} in vitro, which is not observed. Inhibitors can act in concert with a decarbamylated catalytic site by slowing recarbamylation, and activase can help clean up the active sites so they can carbamylate. Reword to better reflect the deactivation mechanism.

RESPONSE

Reworded:

Lines 71-73: “Rubisco is prone to decarbamylation, where a Mg^{2+} ion and CO_2 molecule is not bound to the active site prior to RuBP substrate binding, leading to deactivation and the need for Rca to remove bound RuBP from the active site²⁸.”

9) The acclimation of activation state between cool and warm climate plants also reflects systems where electron transport rate is acclimating and/or adapting. It would seem that this needs to be effectively modelled as well, which is why the lack of explanation regarding the parameterization of June et al is even more problematic.

RESPONSE

We have expanded and presented the June model in greater detail, including the parameters from different species presented in Supplementary Table 3 and the equation itself and explanation of its use in the Methods (Equations 8 and 9).

10)

11. The conclusion paragraph on lines 341 to 356 is a bit strong for the data. e.g. line 344-345 - tough to accept this conclusion when g_m and g_s are not modelled as a function of temperature. The use of "indicates" in line 347 reflects the suggestive nature of the results and contradicts the tone of certainty in the title of the paper and abstract. Line 349 - "drive mounting declines", is based on a flawed analysis as noted above. Line 350 - "speed of adaptation" is unsubstantiated, since this study did not address adaptive speed.

RESPONSE

We have heavily reworded the conclusion and the problematic wording is now removed. We have kept the point that biochemical limitations can explain declines in photosynthesis above the T_{opt} as that is what our results imply. But this is followed by a caveat that the intrinsic limitation by A_c and A_r would also be overlaid with water deficit implications on photosynthesis performance:

Lines 349-353: “Finally, we demonstrated that neither CO_2 substrate supply limitation nor photorespiratory CO_2 loss was needed to explain high temperature-induced decreases in A_n . However, many future heatwaves are likely to coincide with drought, and drought will reduce CO_2 conductance and increase photorespiratory CO_2 loss, exacerbating the stress caused by Rubisco deactivation and declines in J .”

In sum, further model development and effective use of their model to predict

clear examples where activase is known to show control would be a more robust approach. Perhaps avoid conclusions regarding activase being a general limitation until other limitations can be definitively ruled out, both experimentally and in theory. This is a long-standing expectation in photosynthetic limitation research that we all have had to adhere to lest we face unpleasant assessments by colleagues.

RESPONSE

Very good point. This is addressed by the reorientation of manuscript towards accurate prediction of A_n loss with temperature through accounting for activation (or J limitations), not ascribing the limitation to loss of Rca function explicitly.

Sorry to be a downer, but I think you can have a much more powerful study if you build a more comprehensive model and evaluate it with some specific empirical tests of its predictions, as suggested above. - Rowan Sage

RESPONSE

Thank you, you make valid and understandable points. In our revisions, we believe we have addressed your fundamental concerns and the study is much improved because of it.

Reviewer #2:

Remarks to the Author:

I am satisfied with the way the authors responded to my comments. However, I note that supplemental figure 1 could be improved. The panel for K_o the units should be kPa, not KPa. K is reserved for Kelvin and k means 1000, despite what the molecular biologists do. Third line on the legend there is a The that should be the. Finally, the dashes for Galmes et al in vivo data are hard to see. These look like solid lines on my copy.

I would make two other observations as long as I am typing. 1. Limitation is used throughout but I like to distinguish between limitation and regulation. Once something is limited there is no flexibility in one direction. Something that is regulating is setting the pace and the system will behave as though under control of the regulator, but the system will retain flexibility. 2. The cost of photorespiration is invoked in a couple of places. My thinking is that we don't really know the cost of producing 2-phosphoglycolate at high rates. It may be desirable to limit rubisco carbamylation at high temperature so that 2-phosphoglycolate production and accumulation does not go too high for reasons independent of the costs of the reactions involved in metabolizing the 2-PG.

Tom Sharkey

Reviewer #3:

Remarks to the Author:

The authors have done an effective job responding to my comments. I have no further concerns. The study is robust and important and will be well received by the photosynthesis community.

REVIEWERS' COMMENTS

Reviewer #1 (Remarks to the Author):

The authors have carefully and thoroughly addressed my concerns, and those of the other reviewers. They have revised the manuscript significantly and show evidence supporting co-regulation of Rubisco activity and electron transport, and co-limitation of photosynthesis at ambient and elevated temperatures.

My only comment for their consideration would be to use caution with the statement in lines 249-252, as some authors would argue against an active leaf cooling mechanism (e.g. Still et al 2022 PNAS <https://www.pnas.org/doi/abs/10.1073/pnas.2205682119>).

Thank you for pointing out this reference. We have now cited this report (**lines 252-254**) and make an additional caveat that in natural sunlit canopies there is limited evidence of transpiration cooling of leaves below air temperature likely due to the radiative heating and other biophysical factors.

Thanks for the thorough revision and response to comments raised on the first version.

Reviewer #2 (Remarks to the Author):

I am satisfied with the way the authors responded to my comments. However, I note that supplemental figure 1 could be improved. The panel for K_o the units should be kPa, not KPa. K is reserved for Kelvin and k means 1000, despite what the molecular biologists do. Third line on the legend there is a The that should be the. Finally, the dashes for Galmes et al in vivo data are hard to see. These look like solid lines on my copy.

We have corrected to kPa, have changed The to the, and made the lines more distinguishable in Supplementary Figures 1 and 2.

I would make two other observations as long as I am typing. 1. Limitation is used throughout but I like to distinguish between limitation and regulation. Once something is limited there is no flexibility in one direction. Something that is regulating is setting the pace and the system will behave as though under control of the regulator, but the system will retain flexibility.

We have modified a sentence in the discussion (**lines 311-313**) to emphasis this point, that the downregulation in both Rubisco activation and J may be due to both becoming limited (temperature dependent loss of function), one downregulating to the other becoming limited, or a combination of both. We use the word limited more broadly throughout the paper to mean a reduction in A_n with rising temperature which occurs for both Rubisco activation and J irrespective of whether it is due to regulation or direct physical limitation.

2. The cost of photorespiration is invoked in a couple of places. My thinking is that we don't really know the cost of producing 2-phosphoglycolate at high rates. It may be desirable to limit rubisco carbamylation at high temperature so that 2-phosphoglycolate production and accumulation does not go too high for reasons independent of the costs of the reactions involved in metabolizing the 2-PG.

We have added this point to the manuscript (**lines 276-277**) to make it clear that we are judging the cost of photorespiration based on its CO₂ fixation cost to photosynthesis which potentially may not account for other costs associated with 2-phosphoglycolate metabolism.

Tom Sharkey

Reviewer #3 (Remarks to the Author):

The authors have done an effective job responding to my comments. I have no further concerns. The study is robust and important and will be well received by the photosynthesis community.

Rowan Sage